# Generalization Bounds for Kolmogorov-Arnold Networks (KANs) and Enhanced KANs with Lower Lipschitz Complexity

**Pengqi Li**   **Lizhong Ding**[*]   **Jiarun Fu**   **Chunhui Zhang**   **Ye Yuan**   **Guoren Wang**

School of Computer Science and Technology, Beijing Institute of Technology

{pengqi.li, jrfu, zhangch, yuan-ye}@bit.edu.cn

lizhong.ding@outlook.com   wanggrbit@126.com

## Abstract

Kolmogorov-Arnold Networks (KANs) have demonstrated remarkable expressive capacity and predictive power in symbolic learning. However, existing generalization errors of KANs primarily focus on approximation errors while neglecting estimation errors, leading to a suboptimal bias-variance trade-off and poor generalization performance. Meanwhile, the unclear generalization mechanism hinders the design of more effective KANs. As the authors of KANs highlighted, they *"would like to explore ways to restrict KANs' hypothesis space so that they can achieve good performance."* To address these challenges, we explore the generalization mechanism of KANs and design more effective KANs with lower model complexity and better generalization. We define *Lipschitz complexity* as the first structural measure for deep functions represented by KANs and derive novel generalization bounds based on *Lipschitz complexity*, establishing a theoretical foundation for understanding their generalization behavior. To reduce *Lipschitz complexity* and boost the generalization mechanism of KANs, we propose Lipschitz-Enhanced KANs (**LipKANs**) by integrating the Lip layers and pioneering the $L_{1.5}$-regularization, contributing to tighter generalization bounds. Empirical experiments validate that the proposed LipKANs enhance the generalization mechanism of KANs when modeling complex distributions. We hope our theoretical insights and proposed LipKANs lay a foundation for the future development of KANs.

## 1 Introduction

KANs represent multivariate functions as layer-wise iterative compositions of univariate continuous functions [Liu et al., 2025], indicating the strong expressive capacity to model complex distributions and possessing faster neural scaling laws. The fundamental difference between KANs and existing neural network architectures [He et al., 2016, Vaswani, 2017, Brown et al., 2020] is that KANs replace the single activation in each layer with a family of activations. KANs demonstrate strong accuracy and interpretability on AI + Science tasks [Shukla et al., 2024, Wang et al., 2025b], while also achieving promising results in socio-economic tasks [Xu et al., 2024b], engineering-related tasks [Mubarak et al., 2024, Kundu et al., 2024, Bresson et al., 2024, Carlo et al., 2024], and interpretability-driven tasks [Galitsky, 2024]. While KANs demonstrate strong approximation capabilities and excel in symbolic learning, their generalization does not show a significant advantage over existing models [Yu et al., 2024, Zheng et al., 2025]. We suggest that the suboptimal generalization performance of KANs arises from an increased hypothesis space complexity, which introduces an approximation-estimation error trade-off: while lowering approximation error, it elevates sample complexity, thereby raising

---

[*]Corresponding Author

estimation error and ultimately harming generalization. Parameter count serves as an insufficient proxy for hypothesis space complexity in KANs, and naively reducing parameter counts fails to improve generalization. Consequently, understanding KANs' generalization mechanisms and developing complexity measures beyond parameter count are critical for enhancing their applicability and advancing more effective KANs-based models.

Existing studies on the generalization of KANs primarily focus on bounding approximation error. For instance, Ataei et al. [2025] derive a more general approximation result extending beyond depth-2 Kolmogorov-Arnold representations, and Qiu et al. [2024] provide distance-aware error bounds for KANs. Wang et al. [2025a] rigorously establish that KANs are at least as expressive as MLPs by showing that any MLP can be reparameterized as a KAN with degree $k$ splines which is only slightly larger and KANs can also be represented by MLPs, implying that KANs with large grid sizes may be more efficient in approximating certain classes of functions. However, these studies largely overlook the impact of estimation error, limiting their contribution to improving KANs' generalization performance. To our knowledge, the only existing generalization bounds [Zhang and Zhou, 2024] provide a foundation for KANs but overlook their structural complexity, yielding limited insight into their generalization mechanisms and offering little guidance for more effective models.

In this paper, we derive the generalization bounds for KANs and propose more effective Lipschitz-Enhanced KANs (**LipKANs**), with lower complexity and better generalization performance. We reformulate KANs in Eq. (1) by treating each layer as a combination of activation functions and a linear transformation, highlighting that the core of KANs' expressive capacity arises from the family of activations in each layer. We define a structural measure, *Lipschitz complexity*, for the function family represented by KANs and, based on this, directly evaluate the generalization error of KANs by deriving their generalization bounds in Theorem 4.5. These bounds: (a) scale with *Lipschitz complexity* of KANs; (b) have no dependence on combinatorial parameters (e.g. number of layers or cardinality of activations family) outside of log factors; (c) regard the previous generalization bounds for MLPs as special cases; (d) accommodate the selection of various activations. We show that adopting activations with lower *Lipschitz complexity* results in tighter bounds, while incorporating complexity-related regularization during training helps trade off bias and variance. Leveraging the theoretical analyses, we design LipKANs, which incorporate the Lip layers and replace the $L_1$-regularized loss [Liu et al., 2025] with the $L_{1.5}$-regularized loss, where the Lip layers ensure the low *Lipschitz complexity*, and regularization helps balance the bias-variance trade-off. Significantly, we conclude that LipKANs lead to better generalization both theoretically and empirically.

**Contributions**   This work establishes the generalization bounds for KANs and proposes LipKANs, with an enhanced generalization mechanism. The central contributions are as follows:

- **Theoretical Foundation**: We are the first to rigorously define Lipschitz complexity as a measure of the structural complexity of deep functions represented by KANs, and to derive complexity-dependent generalization bounds.

- **Architectural Innovation**: We introduce a generalization-driven paradigm for model design that establishes clearer generalization mechanisms. Specifically, we introduce LipKANs, which insert Lip layers and apply $L_{1.5}$-regularization to explicitly bound network complexity.

- **Broad Adaptability**: We evaluate LipKANs on vision, text, and multimodal benchmarks, demonstrating that they consistently improve existing KANs, with significant accuracy gains when modeling complex distributions.

## 2   Related works

### 2.1   Generalization of Deep Models

Generalization error is defined as the gap between the expected risk (expected loss over the data distribution) and the empirical risk (observed loss on the training set). Therefore, a smaller generalization error indicates better model generalization. A generalization bound is a theoretical upper bound on the generalization error, taking the form: expected risk $\leq$ empirical risk + complexity + confidence, where the complexity term reflects the expressiveness of the hypothesis set, and the confidence term quantifies how likely the bound holds.

Most of the studies on the generalization of deep models are based on generalization bounds [Jiang et al., 2019, Dziugaite et al., 2020]. Depending on the selection of the complexity term, they can be classified into: VC dimension bounds [Shalev-Shwartz and Ben-David, 2014, Neal et al., 2018, Ding et al., 2020b, Yang et al., 2023], Rademacher complexity bounds [Bartlett and Mendelson, 2002, Sachs et al., 2023], margin bounds [Bartlett et al., 2017, Neyshabur et al., 2017, Ding et al., 2018, Barron and Klusowski, 2019], NTK-based bounds [Arora et al., 2019], PAC-Bayes bounds [Eringis et al., 2024], and uniform stability bounds [Hardt et al., 2016, Mou et al., 2018, Attia and Koren, 2022]. However, many generalization bounds fail to reveal the underlying generalization mechanism of the models, making them less helpful for guiding model enhancements.

### 2.2 Kolmogorov-Arnold Networks (KANs)

Since the emergence of KANs [Liu et al., 2025], future researches can be categorized into four directions: (I) Explore the applications of KANs in fields such as graphs [Bresson et al., 2024, Carlo et al., 2024, Kiamari et al., 2024, Zhang and Zhang, 2024], computer vision [Azam and Akhtar, 2024, Li et al., 2024, Cheon, 2024, Seydi et al., 2024], kernel learning [Zinage et al., 2024] and quantum science [Kundu et al., 2024, Ahmed and Sifat, 2024]. These studies have expanded the application scope of KANs, but lack theoretical analysis [Liu et al., 2019]. (II) Explore various activation functions, including Gaussian radial basis function [Li, 2024], Fourier series [Xu et al., 2024a], Chebyshev polynomials [SS and R, 2024], finite basis [Howard et al., 2024, Ta, 2024], wavelet [Bozorgasl and Chen, 2024, Seydi et al., 2024], Jacobi basis functions [Aghaei, 2024a], polynomial basis functions [Seydi, 2024], and rational functions [Aghaei, 2024b]. These KANs variants propose empirically more effective KANs but lack a theoretical foundation. (III) Combine KANs with models like Convolutional KANs [Bodner et al., 2024], federated KANs [Zeydan et al., 2024] and PowerMLP [Qiu et al., 2024]. These works blend KANs with other deep networks, though they may sacrifice KANs' inherent expressive capacity. (IV) Research the approximation error of KANs [Qiu et al., 2024, Ataei et al., 2025]. Our study on generalization error overlaps with this area, suggesting that focusing only on approximation error may reduce the model's reliability and stability when handling real-world data, thus limiting its practical utility. The trade-off between estimation and approximation errors is crucial for improving generalization performance.

## 3 Notations and Preliminaries

In the supervised learning scenario, we assume that data points $(\boldsymbol{x}_i, y_i) \in \mathcal{X} \times \mathcal{Y}$ for $i \in [n]$ are drawn independently from a distribution $\mathbb{P}$. The sample is denoted as $S = \{(\boldsymbol{x}_1, y_1), (\boldsymbol{x}_2, y_2), \ldots, (\boldsymbol{x}_n, y_n)\} \sim \mathbb{P}^n$. The input sample can be represented in matrix form as $X = (\boldsymbol{x}_1, \boldsymbol{x}_2, \ldots, \boldsymbol{x}_n)^{\mathrm{T}}$. Given a loss function $\mathcal{L}(\cdot, \cdot) : \mathcal{Y} \times \mathcal{Y} \to \mathbb{R}_+$ and a hypothesis space $\mathcal{H}$, the expected loss of $f \in \mathcal{H}$ is defined as $R(f) = \mathbb{E}_{(\boldsymbol{x}, y) \sim \mathbb{P}}[\mathcal{L}(f(\boldsymbol{x}), y)]$ and the empirical loss of $f \in \mathcal{H}$ on the set $S$ is given by $\hat{R}(f) = \frac{1}{n} \sum_{i=1}^{n} \mathcal{L}(f(\mathrm{x}_i), y_i)$. In Section 4, we aim to derive upper bounds for $R(f)$ that hold for all $f \in \mathcal{H}$ with high probability. These bounds will depend on both $\hat{R}(f)$ and the complexity term. In Appendix C.1, we formally define the norms of vectors (e.g., $\boldsymbol{x_i}$), matrices (e.g., the input matrix $X$) and three-dimensional tensors (e.g., the weight tensor $\mathbf{W}_i$ of KANs). Our analysis focuses on sub-Gaussian distributions $\mathbb{P}$; formal definitions of sub-Gaussian random variables and vectors are provided in Appendix C.2.

**KANs.** It was established by KART [Kolmogorov, 1961] that, for a continuous $f : [0, 1]^n \to \mathbb{R}^n$, there exist continuous univariate functions $\Phi_q : \mathbb{R} \to \mathbb{R}$ and $\phi_{q,p} : [0, 1] \to \mathbb{R}$ such that:

$$f(\mathrm{x}) = f(x_1, \cdots, x_n) = \sum_{q=1}^{2n+1} \Phi_q \left( \sum_{p=1}^{n} \phi_{q,p}(x_p) \right).$$

To expand the family of functions that a two-layer KAN can represent, it is generalized to be both wider and deeper, and is parameterized by 1D functions using B-spline basis functions [Liu et al., 2025]. The underlying idea is that when using a two-layer KAN to represent functions, some of 1D functions may be difficult to learn, while these functions become learnable when the network is sufficiently deep. For example, $f(x_1, x_2, x_3, x_4) = \exp\left(\sin\left(x_1^2 + x_2^2\right) + \sin\left(x_3^2 + x_4^2\right)\right)$ cannot be smoothly represented by a two-layer KAN but can be smoothly composited by a three-layer KAN with the width $[4, 2, 1, 1]$. We define the activations family of cardinality $G$ as $\Sigma = \{\sigma_k : \mathbb{R} \to \mathbb{R}, k \in$

$[G]\}$. Generally, we consider a KAN of depth $L$ and width $[d_1, d_2, \ldots, d_L]$, where the $l$-th layer is defined by the vector-valued map $\Psi_l : \mathbb{R}^{d_{l-1}} \to \mathbb{R}^{d_l}$ with $\Psi_l(\mathbf{x}) = (\psi_{l,1}(\mathbf{x}), \ldots, \psi_{l,d_l}(\mathbf{x}))$ and each neuron given by $\psi_{l,i}(\mathbf{x}) = \sum_{j=1}^{d_{l-1}} \sum_{k=1}^{G} w_{i,j,k}\, \sigma_k(x_j)$, where in $w_{i,j,k}$, $i$ indexes the neuron in the next layer, $j$ refers to the neuron in the current layer, and $k$ corresponds to the basis function from $\Sigma$.

**Lipschitz Property.** Let $(X, d_X)$ and $(Y, d_Y)$ be metric spaces. A function $f : X \to Y$ is called *Lipschitz* if there exists $L \in \mathbb{R}$, such that $d_Y(f(u), f(v)) \leq L \cdot d_X(u, v), \forall u, v \in X$. The infimum of all $L$ in this definition is called the Lipschitz norm of $f$ and is denoted $\|f\|_{\text{Lip}}$. As for binary functions (such as loss functions $\mathcal{L}(\cdot, \cdot) : \mathcal{Y} \times \mathcal{Y} \to \mathbb{R}_+$), we define their Lipschitz norm by fixing the parameter at one position, represented as a function $\rho(y) := \|\mathcal{L}(\cdot, y)\|_{\text{Lip}}$. The Lipschitz property of various function families is discussed in Appendix C.3.

## 4 Generalization Analysis

In this section, we provide a theoretical framework for understanding a model's ability to generalize from training data to unseen data, which also plays a crucial role in guiding model selection and regularization strategies. We first formalize KANs in terms of activation layers and linear layers, then we define *Lipschitz complexity* as a measure for functions represented by KANs and derive uniform high-probability generalization bounds, thereby revealing the generalization mechanism of KANs.

### 4.1 Reformulation of KANs

For reformulation, we denote the family of non-linearities $\Sigma$ applied element-wise to $\boldsymbol{x} \in \mathbb{R}^d$ as $\Sigma(\boldsymbol{x}) \in \mathbb{R}^{d \times G}$, where each entry is given by $(\Sigma(\boldsymbol{x}))_{ik} = (\sigma_k(x_i))$. We introduce the weight tensor $\mathbf{W}_l \in \mathbb{R}^{d_l \times d_{l-1} \times G}$ with $(\mathbf{W}_l)_{i,j,k} = w_{i,j,k}$. Using the Einstein summation operation (einsum) "$\circ$" (defined in Appendix C.4), we compactly express $\Psi_l$ as: $\Psi_l(\boldsymbol{x}) = \mathbf{W}_l \circ \Sigma(\boldsymbol{x})$. By iterating $\boldsymbol{x}_l = \mathbf{W}_l \circ \Sigma(\boldsymbol{x}_{l-1})$ with $\boldsymbol{x}_0 = \boldsymbol{x}$, we formally represent a KAN as a composition of linear layers $\mathbf{W}_l$ and non-linearities $\Sigma$ :

**Definition 4.1** (Reformulation of KANs). Given an activation family $\Sigma = \{\sigma_k : \mathbb{R} \to \mathbb{R} \mid k \in [G]\}$, a KAN with depth $L$ and layer widths $[d_0, d_1, \ldots, d_L]$ is defined as a composition of nonlinear activation layers $\Sigma$ and linear layers $\mathcal{W}_1^L = (\mathbf{W}_1, \mathbf{W}_2, \ldots, \mathbf{W}_L)$:

$$\text{KAN}(\boldsymbol{x}) = F_{\mathcal{W}_1^L}(\boldsymbol{x}) = (\mathbf{W}_L \circ \Sigma \circ \mathbf{W}_{L-1} \circ \cdots \circ \mathbf{W}_1 \circ \Sigma)(\boldsymbol{x}). \tag{1}$$

For comparison, an MLP with linear weight matrices $W_l \in \mathbb{R}^{d_l \times d_{l-1}}$ and non-linearity $\sigma$ is expressed:

$$\text{MLP}(\boldsymbol{x}) = (\mathbf{W}_L \circ \sigma \circ \mathbf{W}_{L-1} \circ \cdots \circ \sigma \circ \mathbf{W}_1)(\boldsymbol{x}),$$

where $\sigma : \mathbb{R} \to \mathbb{R}$ is an activation function (e.g., ReLU, sigmoid, or Tanh). Thus, in each layer, KANs replace the single activation in MLPs with a family of activations, contributing to their strong expressive capacity.

### 4.2 Assumptions

For our research subject, KANs, we make the following assumptions:

**Assumption 1** (Sub-Gaussian). We assume that each row $\boldsymbol{x_i} \in \mathbb{R}^d$ of the input data matrix $X$ is an independent and identically distributed sub-Gaussian random vector, with the corresponding sub-Gaussian norm $K := \max_i \|\boldsymbol{x}_i\|_{\varphi_2}$.

Assumption 1 is satisfied by a bounded dataset, and ensures the concentration of the data matrix $\|X\|_2$ with high probability. Indeed, we have $\forall t > 0, \Pr\left(\|\boldsymbol{x}_i\|_2 \geq C(\sqrt{n}K + t)\right) \leq e^{-t^2}$, where $C$ is a constant. Using the concentration bound for the sum of squared sub-Gaussian vectors, we obtain the following lemma:

**Lemma 4.1.** *Under Assumption 1, we have*

$$\Pr\left(\|X\|_2 \geq CK\sqrt{nd\log(1/\delta)}\right) \leq \delta, \tag{2}$$

*where $C$ is a constant.*

Lemma 4.1 demonstrates that, under assumption 1, $\|X\|_2$ concentrates around $\mathcal{O}(K\sqrt{nd})$ with exponential tail decay, ensuring that deviations beyond this bound are extremely unlikely. A detailed discussion of the theoretical extension to other distributions [Ding et al., 2019] is provided in Appendix F.

**Definition 4.2** (Lipschitz Norm of Functions Family). Consider a finite family of functions $\Sigma = \{\sigma_k : \mathbb{R} \to \mathbb{R}, k \in [G]\}$, which corresponds one-to-one with a matrix-valued function $M_\Sigma : \mathbb{R}^d \to \mathbb{R}^{d \times G}$, and for a given vector $\boldsymbol{x} = (x_1, x_2, \cdots, x_d)^{\mathrm{T}} \in \mathbb{R}^d$, the $(i, k)$-th element of the matrix $M_\Sigma(\boldsymbol{x})$ is given by $\sigma_k(x_i)$. The Lipschitz norm of this matrix-valued function is then defined as:

$$\|M_\Sigma\|_{\mathrm{Lip}} := \sqrt{\sum_{k=1}^G \|\sigma_k\|_{\mathrm{Lip}}^2}.$$

For convenience, we write $\|\Sigma\|_{\mathrm{Lip}} := \|M_\Sigma\|_{\mathrm{Lip}}$.

**Assumption 2** (Lipschitz). We assume the family of basis functions $\Sigma$ applied to KANs has bounded Lipschitz norm $\|\Sigma\|_{\mathrm{Lip}} \le C$, with $\|\Sigma\|_{\mathrm{Lip}}$ defined in Decinition 4.2:

To check the consistency of this definition with the previous definition of the Lipschitz norm for a single function, we note that the vector space and matrix space are endowed with the vector 2-norm and the matrix $(2, 2)$-norm, respectively. We cautiously point out that the notation $\|\mathcal{F}\|_{\mathrm{Lip}}$ here involves a slight abuse of symbol, as it does not represent a norm in the formal mathematical sense.

**Assumption 3.** We assume that the loss function $\mathcal{L}(\cdot, \cdot)$ is Lipschitz continuous with respect to its first positional argument, i.e. $\max_{y \in \mathcal{Y}} \rho(y) < \infty$. Further suppose that $\mathcal{L}(\cdot, \cdot) \le M$.

This assumption ensures the Lipschitz continuity of the mapping from the hypothesis space to the loss function space. Both cross-entropy loss ($\rho(y) = 1$) and margin loss ($\rho(y) = \frac{1}{\gamma}$) satisfy this property, as well as $L_p$-loss on a bounded domain.

## 4.3 Generalization Bounds for KANs

The primary goal of this section is to derive uniform high-probability generalization bounds for KANs with auxiliary lemmas (please refer to Appendix A).

We outline the proof by first deriving layer-wise covering number bounds via induction (Lemma 4.2), then establishing tensor covering results for the linear transformations within each layer (Lemma 4.3), and finally obtaining the generalization bounds for KANs using standard generalization analysis techniques (Theorem 4.4).

We begin by decomposing the covering number for multi-layer function compositions into covering numbers for layer-wise functions. Our techniques are similar to Lemma A.7 in [Bartlett et al., 2017].

**Lemma 4.2.** Let $(\epsilon_1, \ldots, \epsilon_L)$ be given. Under Assumption 1 and 2, each $\sigma_k$ is Lipschitz continuous with $\rho_k := \|\sigma_k\|_{Lip}$. Let $\rho := (\rho_1, \ldots, \rho_G)$. Suppose the tensors $\mathcal{W}_1^L = (\mathbf{W}_1, \ldots, \mathbf{W}_L)$ lie within $\mathcal{B}_1 \times \cdots \times \mathcal{B}_L$ where $\mathcal{B}_i$ are arbitrary classes with the property that each $\mathbf{W}_i \in \mathcal{B}_i$ has $\|\mathbf{W}_i\|_\sigma \le c_i$. Then, letting $\tau_1 = \epsilon_1$ and $\tau_l = \sum_{i=1}^l (\prod_{j=i+1}^l c_j \|\rho\|_2) \epsilon_i$, the neural net images $\mathcal{H}_X := \left\{ F_{\mathcal{W}_1^L}(X) : \mathcal{W}_1^L \in \mathcal{B}_1 \times \cdots \times \mathcal{B}_L \right\}$ have a covering number bound:

$$\mathcal{N}\left(\mathcal{H}_X, \tau_L, \|\cdot\|_2\right) \le \prod_{i=1}^L \sup_{\substack{(\mathbf{W}_1, \ldots, \mathbf{W}_{i-1}) \\ \forall j < i \cdot \mathbf{W}_j \in \mathcal{B}_j}} \mathcal{N}\left(\left\{\mathbf{W}_i \circ \Sigma\left(F_{\mathcal{W}_1^{i-1}}(X)\right) : \mathbf{W}_i \in \mathcal{B}_i\right\}, \epsilon_i, \|\cdot\|_2\right). \tag{3}$$

The complete proof is provided in Appendix E.1. Lemma A.2 shows that it suffices to study the covering number of the $l$-th layer of the KAN, given that the first $l - 1$ layers are fixed.

To establish the tensor covering bounds for the linear transformation, we demonstrate Lemma 4.3 that considers $\mathbf{W} \circ \Sigma(\mathbf{X})$, where $\mathbf{W}$ represents the weight tensor for a given layer, and $X$ denotes the data passed through all preceding layers.

**Lemma 4.3.** *Let conjugate exponents $(p, q)$ and $(u, v)$ be given with $p \leq 2$ as well as positive reals $(a, b, \epsilon)$ and positive integer $m$. Let data $X$ be given with $\|\Sigma(X)\|_p \leq b$. Then*

$$\ln \mathcal{N}\left(\left\{\mathbf{W} \circ \Sigma(X) : \mathbf{W} \in \mathbb{R}^{d' \times d \times G}, \|\mathbf{W}\|_{q,q,v} \leq a\right\}, \epsilon, \|\cdot\|_2\right) \leq \left\lceil \frac{a^2 b^2 d'^{\frac{2}{u}}}{\epsilon^2} \right\rceil \ln(2dd'G),$$

$$(4)$$

*where $\lceil x \rceil$ returns the smallest integer greater than or equal to $x$.*

The complete proof is provided in Appendix E.2. Now we define *Lipschitz complexity* as a measure for the functions represented by KANs.

**Definition 4.3** (Lipschitz Complexity). The *Lipschitz complexity* $R_{\mathcal{W}_1^L}$ of a KAN $F_{\mathcal{W}_1^L}$ with weights $\mathcal{W}_1^L = (\mathbf{W}_1, \mathbf{W}_2, \dots, \mathbf{W}_L)$ is defined as:

$$R_{\mathcal{W}_1^L} := \left( \|\Sigma\|_{\text{Lip}}^L \prod_{i=1}^{L} \|\mathbf{W}_i\|_\sigma \right) \left( \sum_{i=1}^{L} \|\mathbf{W}_i^{\text{T}}\|_{2,2,1}^{2/3} \right)^{\frac{3}{2}},$$

where $\mathbf{W}_i^{\text{T}}$ refers to permuting the 3D tensor $\mathbf{W}_i$ from (dim1, dim2, dim3) to (dim2, dim3, dim1).

**Remark on Lipschitz Complexity.** Lipschitz Complexity acts as a complexity for hypothesis space [Mohri, 2018, Ding et al., 2020a] and is indeed distinct from algorithmic complexity. While the formal definition of Lipschitz complexity looks highly structured, it actually systematically accounts for both intuitive factors (e.g., network depth $L$ and spectral weight norm $\|\mathbf{W}_i\|_\sigma$) and non-intuitive factors (e.g., smoothness of the basis family $\|\Sigma\|_{\text{Lip}}$ and specific (2,2,1) weight norm $\|\mathbf{W}_i^{\text{T}}\|_{2,2,1}$) that characterize KANs. Besides, the logarithmic dependence of generalization bounds on width $d$ and basis family size $G$ ensures their marginal impact on complexity compared to the dominant term $R_{\mathcal{W}_1^L}$. Finally, under Assumptions 1 & 3, we will strictly bound the leftover terms by constants. These designs make Lipschitz complexity a sufficient measure – higher $R_{\mathcal{W}_1^L}$ makes generalization more challenging, whereas lower $R_{\mathcal{W}_1^L}$ facilitates generalization.

With the above results, we can provide a bound on the covering number of $\mathcal{H}_X$, specializing to the particular case of Lemma 4.3, where we take $p = q = 2$, $u = \infty$ and $v = 1$.

**Theorem 4.4** (Full-Network Covering Bounds). *Under Assumption 2, we can bound*

$$\log \mathcal{N}\left(\mathcal{H}_X, \epsilon, \|\cdot\|_2\right) \leq \frac{R_{\mathcal{W}_1^L}^2 \log(2\tilde{d}^2 G)}{\epsilon^2},$$

$$(5)$$

*where $\tilde{d} := \max_i d_i$.*

The whole proof of Theorem 4.4 is provided in Appendix E.3. It is worth noting that the above bound has no dependence on combinatorial parameters such as the number of nodes and the number of activations outside of the logarithmic factor. Built upon the above results, we now derive bounds on the generalization error $R(F_{\mathcal{W}_1^L})$. The following theorem provides generalization bounds for KANs whose activations is fixed but *Lipschitz complexity* is bounded.

**Theorem 4.5** (Generalization Bounds for KANs). *Under Assumption 2 and 3, let fixed Lipschitz activations $\Sigma = \{\sigma_k \mid k \in [G]\}$ and weight tensors $\mathcal{W}_1^L = \{\mathbf{W}_1, \mathbf{W}_2, \dots, \mathbf{W}_L\}$ be given. Then for $(\boldsymbol{x}, y), (\boldsymbol{x}_1, y_1), \dots, (\boldsymbol{x}_n, y_n)$ drawn i.i.d. from distribution $\mathbb{P}$, we have with probability greater than $1 - \epsilon$,*

$$R(F_{\mathcal{W}_1^L}) - \hat{R}(F_{\mathcal{W}_1^L}) \leq \mathcal{O}\left( \frac{\|X\|_2 R_{\mathcal{W}_1^L} \max_i |\rho(y_i)|}{n} \log(\tilde{d}^2 G) + \sqrt{\frac{1/\epsilon}{n}} \right).$$

$$(6)$$

**Remark on Generalization Bounds.** We acknowledge that the above bounds may not achieve optimal tightness. Nevertheless, our bounds reveal KANs' generalization mechanisms through Lipschitz complexity, provide theoretical guidance for model design, and predict empirical behaviors like the $\mathcal{O}(n^{-\frac{1}{2}})$ error decay. While our current generalization bound does not make inter-layer dependencies explicit, it already captures layer-wise alignment through the operator norm products $\prod_{i=1}^{L} \|\mathbf{W}_i\|_\sigma$, which originate from the covering number bound of the entire network in Lemma 4.2.

This implicitly reflects structural relationships across layers. We agree that making these dependencies more explicit could tighten the bound, and we consider this a valuable direction for future refinement.

Some studies [Qiu et al., 2024, Wang et al., 2025a] theoretically demonstrate stronger expressive power of KANs than MLPs on bounded domains. Here, we remark that a special case of generalization bounds corresponds precisely to the generalization bounds of the MLPs [Bartlett et al., 2017]. When the activations family $\Sigma$ contains only a single activation function, we have $G = 1$, then the result coincides with the classical margin bound in [Bartlett et al., 2017].

Based on Assumption 1, which states that the distribution $\mathbb{P}$ satisfies the sub-Gaussian property, we can bound $\|X\|_2$ in Eq. (6) with high probability, leading to more applicable bounds in Corollary E.1 (See Appendix E.5). Based on the generalization bounds of KANs in terms of *Lipschitz complexity*, we gain the insight that adopting activations with lower *Lipschitz complexity* results in tighter bounds, while incorporating $\|\mathbf{W}_i^T\|_{2,2,1}$-related regularization during training helps trade off bias and variance in Eq. (6). Building upon this, in the next section, we propose LipKANs, which enjoy tighter generalization bounds.

# 5 LipKANs: Generalization-Driven Models

With our efforts in Section 4 , in this section, we introduce LipKANs, which incorporate the Lip layer and replace the $L_1$-regularized loss [Liu et al., 2025] with the $L_{1.5}$-regularized loss, where the Lip layer ensures the low *Lipschitz complexity*, and regularization helps balance the trade-off between bias and variance.

We first introduce the structure of LipKANs. From Theorem 4.5, it follows that adopting function families with greater Lipschitz smoothness leads to tighter bounds, while incorporating complexity-related regularization during training alleviates training costs. Therefore, we design the Lip layer in each KANs layer between each activation layer $\Sigma$ and linear layer $\mathbf{W}_i$ to reduce the Lipschitz norm of the basis function family. The fixed Lipschitz function $\ell : \mathbb{R} \to \mathbb{R}$ satisfies $\|\ell\|_{\text{Lip}} \leq 1$ (e.g., $\tanh$, Sigmoid, ReLU). Given an activations family $\Sigma = \{\sigma_k : \mathbb{R} \to \mathbb{R} \mid k \in [G]\}$, a LipKAN with depth $L$ and layer widths $[d_0, d_1, \ldots, d_L]$ is defined as a composition of nonlinear activation layers $\Sigma$, Lip layers $l$ and linear layers $\mathcal{W}_1^L = (\mathbf{W}_1, \mathbf{W}_2, \ldots, \mathbf{W}_L)$:

$$\text{LipKAN}(\boldsymbol{x}) = F_{\mathcal{W}_1^L, \ell}(\boldsymbol{x}) = (\mathbf{W}_L \circ \ell \circ \Sigma \circ \mathbf{W}_{L-1} \circ \cdots \circ \mathbf{W}_1 \circ \ell \circ \Sigma)(\boldsymbol{x}), \qquad (7)$$

where $\ell$ acts element-wise. Next, we define the $L_{1.5}$-regularized loss to address the imbalance between

---

**Algorithm 1** Generalization-Driven LipKANs with $L_{1.5}$-Regularization

**Require:** Sample $S = \{(\boldsymbol{x}_i, y_i)\}_{i=1}^n$, Activations family $\Sigma$, Function $\ell$ with $\|\ell\|_{\text{Lip}} \leq 1$, Learning rate $\eta$, Regularization coefficient $\lambda$.
**Ensure:** Model weights $\mathcal{W}_1^L = (\mathbf{W}_1, \ldots, \mathbf{W}_L)$.
 1: Initialize weight tensors $\mathcal{W}_1^L$.               ▷ with Kaiming initialization [He et al., 2016].
 2: Initialize AdamW optimizer with parameters:
     $\theta_0 = \{\mathcal{W}_1^L\}, \eta, \lambda_{\text{wd}}$ (weight decay), $\beta_1 = 0.9, \beta_2 = 0.999$.
 3: **for** each epoch $t = 0$ to $T - 1$ **do**
 4:    **for** each batch $(\boldsymbol{x}_i, y_i) \in S$ **do**
 5:       $\boldsymbol{z}^{(0)} \leftarrow \boldsymbol{x}_i$
 6:       **for** $j = 1$ to $L$ **do**                              ▷ Forward propagation.
 7:         $\boldsymbol{z}^{(j)} \leftarrow \mathbf{W}_j \circ \ell \circ \Sigma(\boldsymbol{z}^{(j-1)})$   ▷ Leading to lower $\|\Sigma\|_{\text{Lip}}$ and $R_{\mathcal{W}_1^L}$ (Definition 4.3).
 8:       **end for**
 9:       $F_{\mathcal{W}_1^L, \ell}(\boldsymbol{x}_i) \leftarrow \boldsymbol{z}^{(L)}$
10:       $\mathcal{L} = \mathcal{L}_{\text{entropy}} + \lambda \sum_{j=1}^L \|\mathbf{W}_j^T\|_{2,2,1}$       ▷ $L_{1.5}$ enhances generalization (Corollary 5.1).
11:       Update: $\theta_{t+1} = \text{AdamW}(\theta_t, \mathcal{L}, \eta, \lambda_{\text{wd}})$          ▷ Employ the AdamW optimizer.
12:    **end for**
13: **end for**
14: **return** $\mathcal{W}_1^{L\star} \leftarrow \theta_T$
15: // Integrate Lip layers and design $L_{1.5}$-regularization optimized via AdamW to enhance generalization and training stability.

---

bias and variance. As illustrated in Algorithm 1, we note that the design of $L_{1.5}$-loss originates from generalization bounds (Theorem 4.5), which regularize $\|\mathbf{W}_i^{\mathrm{T}}\|_{2,2,1}$ in $R_{\mathcal{W}_1^L}$, leading to an improved bias-variance trade-off (Corollary E.1). Given dataset $S = \{(\boldsymbol{x}_1, y_1), (\boldsymbol{x}_2, y_2), \ldots, (\boldsymbol{x}_n, y_n)\} \sim \mathbb{P}^n$, basis function family $\Sigma$ and weight tensors $\mathcal{W}_1^L = \{\mathbf{W}_1, \mathbf{W}_2, \ldots, \mathbf{W}_L\}$, we design the $L_{1.5}$-regularized loss to replace the $L_1$-regularized loss in [Liu et al., 2025] based on Theorem 4.5 as:

$$\mathcal{L}_{1.5}(S; \mathcal{W}_1^L, \ell) = \mathcal{L}_{\mathrm{ce}}(S; \mathcal{W}_1^L, \ell) + \lambda \sum_{i=1}^{L} \|\mathbf{W}_i^{\mathrm{T}}\|_{2,2,1},$$

where $\mathcal{L}_{\mathrm{ce}}$ is the cross-entropy loss and $\lambda$ controls the scale of regularization. Then we employ AdmaW optimizer [Loshchilov and Hutter, 2017] with the optimization problem of LipKANs designed as: $\mathcal{W}_1^{L\star} = \arg\min_{\mathcal{W}_1^L} \mathcal{L}_{1.5}(S; \mathcal{W}_1^L, \ell)$. We base model selection on theoretical foundations, but empirically compare regularization norms $(L_1, L_{1.5})$ in Section 6.2 and discuss their theoretical implications in Appendix B. Both KANs [Liu et al., 2025] and Efficient-KANs [2] utilize $L_1$-regularization with the penalty term $\sum_{i=1}^{L} \|\mathbf{W}_i\|_1$ to promote sparsity and interpretability. In contrast, we adopt $L_{1.5}$-regularization, which enhances generalization performance. A promising future direction is to explore integrating $L_1, L_{1.5}$, and the entropy regularization to leverage their complementary advantages. We analyze the cost of Lip layers and $L_{1.5}$-regularization in Appendix D.3.

**Remark on LipKANs.** Both components are principled and arise naturally from our formulation of Lipschitz complexity (Definition 4.2). Specifically, the Lip Layer constrains the smoothness of the basis family through the Lipschitz norm of $\Sigma$, while the $L_{1.5}$-regularization penalizes the structured $(2,2,1)$ norm of the linear weight tensors $\mathbf{W}$. These two components regulate orthogonal axes of Lipschitz complexity and hence play complementary roles in promoting generalization.

Finally, we prove that LipKANs exhibit lower Lipschitz complexity, which tightens the generalization bounds and theoretically enhances the generalization performance.

**Corollary 5.1** (Generalization Bounds for LipKANs). *Under Assumption 1, 2 and 3, let fixed Lipschitz activations $\Sigma = \{\sigma_k : k \in [G]\}$, weight tensor $\mathcal{W}_1^L = \{\mathbf{W}_1, \mathbf{W}_2, \ldots, \mathbf{W}_L\}$ and Lipschitz function $l : \mathbb{R} \to \mathbb{R}$ with $\|l\|_{Lip} \leq 1$. Then for $(\boldsymbol{x}, y), (\boldsymbol{x}_1, y_1), \ldots, (\boldsymbol{x}_n, y_n)$ drawn i.i.d. from a sub-Gaussian distribution $\mathbb{P}$, we have with probability greater than $1 - \epsilon - \delta$,*

$$R(F_{\mathcal{W}_1^L, l}) - \hat{R}(F_{\mathcal{W}_1^L, l}) \leq \mathcal{O}\left( \frac{K R_{\mathcal{W}_1^L, l} \max_i |\rho(y_i)|}{n} \log(\tilde{d}^2 G) \sqrt{\tilde{d} \log\left(\frac{1}{\delta}\right)} + \sqrt{\frac{1/\epsilon}{n}} \right),$$

*where the right-hand side of the inequality is also bounded by the upper bound in Theorem 4.5.*

The proof is deferred to Appendix E.5. Due to Corollary 5.1, adopting activations with lower *Lipschitz complexity* results in tighter generalization bounds, suggesting better generalization performance theoretically. In the next section, we will empirically evaluate the performance of LipKANs.

**Bias-Variance Trade-Off.** To clarify the bias-variance trade-off in our setting, we revisit the generalization bounds in Theorem 4.5 and Corollary 5.1. The Lipschitz complexity term reveals that the model variance—its sensitivity to training data perturbations—is closely related to the structured $(2, 2, 1)$-norm of the weight tensors, which controls the overall magnitude of the spline coefficients. A larger norm enables greater flexibility in fitting the data but also increases the risk of overfitting, thereby contributing to higher variance. In contrast, the bias stems from the approximation capacity of the basis functions (e.g., B-spline, Fourier, and Chebyshev basis). These basis functions provide strong approximation capacity for smooth target functions, enabling the learned model to closely align with the underlying data distribution when appropriately regularized. To this end, the proposed $L_{1.5}$-regularization penalizes the structured norm, reducing variance while preserving expressivity—thereby encouraging a favorable bias-variance trade-off and enhancing generalization.

# 6 Experiments

## 6.1 Deep Learning Tasks

**Baseline.** We choose Original KAN [Liu et al., 2025], Rational-KAN [Yang and Wang, 2024], and RBF-KAN [Ta, 2024] as our baselines. In particular, Rational-KAN and RBF-KAN replace the

---

[2]https://github.com/Blealtan/efficient-kan

B-spline function with their respective basis functions. For all baseline methods, a LayerNorm layer is added before each KAN-Linear layer to enhance training stability. All KANs are set with a grid size of 5, a grid range of $[-10, 10]$, and a hidden layer width of 64. The Lip layer of LipKANs tried using tanh, sigmoid, and ReLU, and selected the best-performing Lip function $\ell$.

**Datasets.** We test the performance of the Lip structures on original KAN, Rational-KAN, and RBF-KAN through benchmark datasets spanning image (MNIST [LeCun et al., 2010], CIFAR-10 [Krizhevsky et al., 2009], CIFAR-100 [Krizhevsky et al., 2009], STL-10 [Coates et al., 2011]), text (AG News [Zhang et al., 2015]), and multimodal domains (AVMNIST [Liang et al., 2021], MIMIC-III [Johnson et al., 2016, Harutyunyan et al., 2019]), confirming LipKANs' robust generalization improvements across broad applications.

All reported accuracy and F1 scores in Table 1 are evaluated on held-out test sets using the officially predefined training/test splits for each dataset, consistent with the evaluation protocols established in prior works [Liu et al., 2025, Zheng et al., 2025].As shown in Table 1, the design of LipKANs demonstrates significantly improved generalization across various KANs when evaluated on complex datasets (SLT-10, CIFAR-100, AG News, and MIMIC-III). In contrast, simpler benchmark datasets (MNIST, CIFAR-10, AVMNIST) demonstrate statistically insignificant performance variation, even exhibiting marginal accuracy degradation in certain cases. This phenomenon stems from the inherently low-complexity distributions captured by KANs on such datasets, rendering hypothesis space complexity reduction unnecessary.

Table 1: Evaluation of KANs and LipKANs on broad domains including CV, NLP, and MM (F1 score (%) for MIMIC-III; accuracy (%) for others) with corresponding 95% confidence intervals computed over five independent runs.

| Model | SLT-10 | MNIST | CIFAR-10 | CIFAR-100 | AG News | AVMNIST | MIMIC-III |
|---|---|---|---|---|---|---|---|
| KAN | $25.94 \pm 0.53$ | $95.62 \pm 0.08$ | $46.01 \pm 0.50$ | $14.77 \pm 0.33$ | $65.58 \pm 0.45$ | $71.91 \pm 0.28$ | $66.39 \pm 0.60$ |
| LipKAN | $38.77 \pm 1.39$ | $97.71 \pm 0.32$ | $50.40 \pm 0.84$ | $20.51 \pm 0.57$ | $68.12 \pm 0.52$ | $72.23 \pm 0.35$ | $70.18 \pm 0.75$ |
| Increase | **+12.83** | **+1.09** | **+4.39** | **+5.74** | **+2.54** | **+0.32** | **+3.79** |
| Rational-KAN | $35.92 \pm 0.85$ | $96.79 \pm 0.15$ | $50.23 \pm 0.62$ | $20.84 \pm 0.42$ | $71.56 \pm 0.38$ | $71.63 \pm 0.31$ | $65.97 \pm 0.55$ |
| LipRational-KAN | $40.67 \pm 1.12$ | $97.01 \pm 0.25$ | $51.12 \pm 0.78$ | $22.79 \pm 0.51$ | $74.98 \pm 0.43$ | $72.42 \pm 0.33$ | $69.12 \pm 0.82$ |
| Increase | **+4.75** | **+0.22** | **+0.89** | **+1.95** | **+3.42** | **+0.79** | **+3.15** |
| RBF-KAN | $37.24 \pm 0.92$ | $96.15 \pm 0.18$ | $47.77 \pm 0.58$ | $18.46 \pm 0.39$ | $62.85 \pm 0.50$ | $70.58 \pm 0.30$ | $67.52 \pm 0.65$ |
| LipRBF-KAN | $40.93 \pm 1.25$ | $96.63 \pm 0.22$ | $49.15 \pm 0.72$ | $20.05 \pm 0.48$ | $66.27 \pm 0.56$ | $71.04 \pm 0.36$ | $74.25 \pm 0.88$ |
| Increase | **+3.69** | **+0.48** | **+1.38** | **+1.59** | **+3.42** | **+0.46** | **+6.73** |

## 6.2 Ablation Study

In this section, we perform ablation experiments to disentangle the contributions of the Lip layer and $L_{1.5}$-regularization. We train original KANs (with default $L_1$ regularization), KANs with $L_{1.5}$-regularization, KANs with Lip Layers, and LipKANs on datasets MNIST, CIFAR10, CIFAR100, and STL10, with training dynamics visualized in Figure 1. Following prior practices, we consistently used $\lambda = 1$ for regularization and $\ell = \tanh$ as the Lipschitz function. Experimental results demonstrated that $L_{1.5}$-regularization played a critical role in enhancing the generalization performance of Lip-KANs. Besides, $L_{1.5}$-regularization shows stronger benefits in complex datasets (e.g., STL-10) due to its ability to regularize high-dimensional weight interactions. Therefore, we strongly recommend adopting $L_{1.5}$-regularization instead of conventional $L_1$-regularization in future research.

To rigorously assess generalization, we follow prior works [Hoffer et al., 2017, Fu et al., 2023] and report three additional metrics in Table 2: Final Gap, Average Gap, and Overfitting Ratio. Notably, on multiple datasets such as CIFAR-10 and MNIST, we observe that KANs + Lip Layers outperform KANs + $L_{1.5}$-regularization on most metrics, suggesting that basis smoothness control is particularly effective in low-noise, low-complexity regimes where architectural inductive bias is more beneficial than weight norm constraints. Conversely, on other datasets (e.g., CIFAR-100, STL-10), the $L_{1.5}$-regularization yields greater improvements, indicating its strength in high-variance regimes where structured norm constraints are better suited to mitigating overfitting. Overall, while both Lip Layers and $L_{1.5}$-regularization aim to control model complexity, they mitigate training variance through different mechanisms, leading to distinct sensitivities to sample-level noise.

Interestingly, some models achieve comparable validation performance yet differ markedly in their generalization gaps. This phenomenon reflects differences in the underlying bias–variance trade-off: models with high training variance may overfit the training data, resulting in deceptively strong validation accuracy but poor generalization. In contrast, methods like those in LipKANs introduce mild bias through regularization while substantially reducing variance, yielding more stable and reliable generalization across datasets. Finally, we analyze the role of activation family size $G$

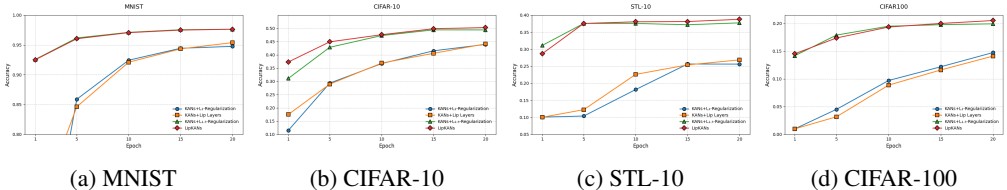

| (a) MNIST | (b) CIFAR-10 | (c) STL-10 | (d) CIFAR-100 |

Figure 1: **Test accuracy across datasets:** (a)-(d) show convergence dynamics of different architectures on four datasets. LipKANs (red diamonds) consistently achieve the highest accuracy with fastest convergence. Comparative analysis reveals $L_{1.5}$-regularization (green triangles) contributes more to generalization improvement than Lipschitz layers (orange squares).

Table 2: Comprehensive ablation analysis of generalization across datasets. Combining both components (i.e., LipKANs) consistently yields the lowest generalization gap and best validation accuracy, indicating their complementary effects.

| Dataset | Model | Final Gap (%) | Avg. Gap (%) | Overfit Ratio (%) | Val. Acc (%) |
|---|---|---|---|---|---|
| CIFAR-10 | KANs ($L_1$-Regularization) | 6.04 | 5.64 | 13.13 | 46.01 |
| | KANs + Lip Layers | 4.45 | 4.92 | 9.47 | 46.99 |
| | KANs + $L_{1.5}$-Regularization | 4.99 | 5.28 | 10.11 | 49.37 |
| | LipKANs | **3.16** | **4.89** | **6.26** | **50.40** |
| CIFAR-100 | KANs ($L_1$-Regularization) | 6.55 | 5.13 | 44.35 | 14.77 |
| | KANs + Lip Layers | 5.84 | 5.10 | 38.98 | 14.98 |
| | KANs + $L_{1.5}$-Regularization | 3.76 | 5.09 | 19.89 | 19.93 |
| | LipKANs | **2.32** | **5.03** | **11.31** | **20.51** |
| MNIST | KANs ($L_1$-Regularization) | 5.52 | 4.92 | 5.77 | 95.62 |
| | KANs + Lip Layers | 3.17 | 4.77 | 3.30 | 96.06 |
| | KANs + $L_{1.5}$-Regularization | 5.13 | 4.81 | 5.27 | 97.26 |
| | LipKANs | **2.55** | **4.70** | **2.60** | **97.71** |
| STL-10 | KANs ($L_1$-Regularization) | 6.93 | 5.27 | 26.72 | 25.94 |
| | KANs + Lip Layers | 4.14 | 5.10 | 15.54 | 26.64 |
| | KANs + $L_{1.5}$-Regularization | 3.95 | 5.06 | 10.47 | 37.73 |
| | LipKANs | **3.31** | **4.98** | **8.53** | **38.77** |

in Appendix D.4. Our analysis shows that the activation family size alone does not determine generalization; rather, the internal structure of the basis functions plays a crucial role.

# 7 Conclusion

In this paper, we establish the foundation for understanding the generalization mechanism of KANs and introduces **LipKANs**, a new network architecture designed to improve generalization. We define *Lipschitz complexity* as the first measure to quantify the structural complexity of deep functions represented by KANs. Based on this measure, we derive complexity-driven generalization bounds, providing a theoretical framework for analyzing KANs' generalization properties. Furthermore, we develop LipKANs, which reduce Lipschitz complexity and enhance the generalization mechanism of KANs. Both theoretical analysis and empirical results confirm that LipKANs achieve tighter generalization bounds and superior performance, paving the way for more effective and theoretically grounded KANs variants. We strongly advocate replacing $L_1$-regularization in KANs with $L_{1.5}$-regularization in future research and hope our bounds establish theoretical foundations for understanding the generalization mechanism of KANs.

## Acknowledgments and Disclosure of Funding

This work was supported by the National Key Research and Development Program of China under Grant 2022YFB2703100, the Joint Funds of the National Natural Science Foundation of China under Grant U22A2099, the National Natural Science Foundation of China under Grant 62376028, the Excellent Young Scientists Fund (Overseas) of the National Natural Science Foundation of China, and the National Key Scientific Instruments and Equipment Development Project under Grant 62427808.

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

# Supplementary Material

## A    Auxiliary Lemmas

This section contains a collection of results that are needed in the proofs, most of which are classical theorems in statistical learning.

Lemma A.1 uses a high-probability upper bound on the local Rademacher complexity[Bartlett et al., 2005] to control the maximal deviation between empirical means and true means of a bounded function, typically a loss function.

**Lemma A.1.** *[(Theorem 2.1 in [Bartlett et al., 2005]] Let $\mathcal{F}$ be a class of functions that map $\mathcal{X}$ into $[a,b]$. Assume that there is some $r \geq 0$ such that for every $f \in \mathcal{F}$, $Var[f(X_i)] \leq r$. Then, for every $x > 0$, with probability at least $1 - 2e^{-x}$ over the data $S = [x_1, x_2, \ldots, x_n]^T$*

$$\sup_{f \in \mathcal{F}} (Pf - P_n f) \leq \inf_{\alpha \in (0,1)} \left( 2\frac{1+\alpha}{1-\alpha} \mathbb{E}_\sigma \mathcal{R}_n \mathcal{F} + \sqrt{\frac{2rx}{n}} + (b-a)\left(\frac{1}{3} + \frac{1}{\alpha} + \frac{1+\alpha}{2\alpha(1-\alpha)}\right)\frac{x}{n} \right). \tag{8}$$

*We denote that $Pf = E_{X \sim P}[f(X)]$ and $P_n f = \frac{1}{n}\sum_{i=1}^n f(x_i)$. For a class $\mathcal{F}$, set $R_n \mathcal{F} = \sup_{f \in \mathcal{F}} R_n f$. Besides, we represent empirical Rademacher complexity as $\mathbb{E}_\sigma \mathcal{R}_n \mathcal{F} = \mathbb{E}_\sigma \left[\frac{1}{n}\sup_{f \in \mathcal{F}} \sigma_i f(x_i)\right]$ where $\sigma_i$ are Rademacher random variables.*

**Corollary A.1.** *Let $\mathcal{F}$ be a class of functions that map $\mathcal{X}$ into $[-M, M]$. Assume that there is some $r \geq 0$ such that for every $f \in \mathcal{F}$, $Var[f(X)] \leq r$. Then for every $\epsilon > 0$, with probability at least $1 - \epsilon$ over the data $S = [x_1, x_2, \ldots, x_n]^T$*

$$\sup_{f \in \mathcal{F}} (Pf - P_n f) \leq 6\mathbb{E}_\sigma \mathcal{R}_n \mathcal{F} + \sqrt{\frac{2r \log(\frac{2}{\epsilon})}{n}} + \frac{32M \log(\frac{2}{\epsilon})}{3n} \tag{9}$$

*Proof.* It suffices to take $\alpha = \frac{1}{2}$, $\epsilon = 2e^{-x}$ and $b - a = 2M$ in Eq. (8) $\qquad \square$

Lemma A.2 is a slight variant of the standard Dudley entropy integral bound on the empirical Rademacher complexity.

**Lemma A.2** (Lemma A.5 in [Bartlett et al., 2017]). *Let $\mathcal{F}$ be a real-valued function class taking values in $[0, M]$ and assume that $\mathbf{0} \in \mathcal{F}$. Then we have*

$$\mathcal{R}(\mathcal{F}(\mathbf{X})) \leq \inf_{a > 0} \left( \frac{4a}{\sqrt{n}} + \frac{12}{n} \int_a^{M\sqrt{n}} \sqrt{\log \mathcal{N}(\mathcal{F}(\mathbf{X}), \epsilon, \|\cdot\|_2)} d\epsilon \right). \tag{10}$$

Lemma A.3 is attributed to Maurey, and applied to bounding the covering number generally.

**Lemma A.3.** *In a Hilbert space $(\mathcal{H}, \|\cdot\|)$, let $U \in \mathcal{H}$ be given with the representation $U = \sum_{l=1}^N a_l W_l$, where $W_l \in \mathcal{H}$ and $a_l \geq 0$. Then, for any positive integer $k$, there exists a choice of nonnegative integers $(k_1, \ldots, k_N)$ such that $\sum_{i=1}^N k_i = k$ and*

$$\left\| U - \frac{\|\mathbf{a}\|_1}{k} \sum_{l=1}^N k_l V_l \right\|^2 \leq \frac{\|\mathbf{a}\|_1}{k} \sum_{l=1}^N a_l \|V_l\|^2 \leq \frac{\|\mathbf{a}\|_1^2}{k} \max_i \|V_i\|^2, \tag{11}$$

*where $\mathbf{a} = (a_1, \ldots, a_N)$ and $\|\mathbf{a}\|_1 = \sum_{l=1}^N a_l$.*

*Proof.* Denote $\beta = \|a\|_1$. Let $W_1, \ldots, W_k$ be $k$ i.i.d. r.v. such that $P(W_1 = \beta V_i) = \frac{a_i}{\beta}$. Assume that $W = \frac{\sum_i W_i}{k}$, then $EW = EW_1 = \sum_i \beta V_i \cdot \frac{a_i}{\beta} = U$.

On the other hand,

$$
\begin{aligned}
E[\|W - U\|^2] &= \frac{1}{k^2} E[\| \sum_i (U - W_i)\|^2] \\
&= \frac{1}{k^2} \left[ E\left[ \sum_i \|U - W_i\|^2 \right] + E\left[ \sum_{i \neq j} \langle U - W_i, U - W_j \rangle \right] \right] \\
&= \frac{1}{k^2} E\left[ \sum_i \|U - W_i\|^2 \right] = \frac{1}{k} E\left[\|U - W_i\|^2\right] = \frac{1}{k}\left( E\left[\|W_1\|^2\right] - \|U\|^2 \right) \quad (12) \\
&\leq \frac{1}{k} E\left[\|W_1\|^2\right] = \frac{1}{k} \sum_i \frac{\alpha_i}{\beta} \beta^2 \|V_i\|^2 = \frac{\beta}{k} \sum_i \alpha_i \|V_i\|^2 \\
&\leq \frac{\beta^2}{k} \max_i \|V_i\|^2
\end{aligned}
$$

Therefore there exists realization $(j_1, \ldots, j_k) \in \{1, \ldots, d\}^k$ such that $\hat{W}_k = \beta V_{j_i}$, $\hat{W} = \frac{\sum \hat{W}_i}{k}$ and $\|U - \hat{W}\| \leq E\left[\|W - U\|^2\right]$. Finally we conclude our result by taking $k_i = \sum_i \mathbf{1}_{[j_l = i]}$. $\square$

# B  Comparsion with Different Regularization Techniques

In this section we discuss the theoretical implications of different regularization techniques, such as standard $L_2$ penalties for ResNet50 [He et al., 2016], $L_1$ penalities for KANs [Liu et al., 2025], dropout for BERT [Devlin et al., 2019] and $L_{1.5}$ penalities in our work.

$L_1$-**regularization**  For MLPs, $L_1$ regularization of linear weights is used to favor sparsity. In KANs, linear weights are replaced by learnable activation functions, so Liu et al. [2025] define the $L_1$ norm of an activation function $\phi$ to be its average magnitude over its $N_p$ input, i.e.

$$
|\phi|_1 \equiv \frac{1}{N_p} \sum_{s=1}^{N_p} \left| \phi\left(x^{(s)}\right) \right|.
$$

Then for a KAN layer $\Phi$ with $n_{\text{in}}$ inputs and $n_{\text{out}}$ outputs, Liu et al. [2025] define the $L_1$ norm of $\Phi$ to be the sum of $L_1$ norms of all activation functions, i.e.,

$$
|\mathbf{\Phi}|_1 \equiv \sum_{i=1}^{n_{\text{in}}} \sum_{j=1}^{n_{\text{out}}} |\phi_{i,j}|_1.
$$

The total training objective $\mathcal{L}_{\text{total}}$ is the prediction loss $\mathcal{L}_{\text{pred}}$ plus $L_1$ regularization of all KAN layers:

$$
\mathcal{L}_{\text{total}} = \mathcal{L}_{\text{pred}} + \lambda \sum_{l=0}^{L-1} |\Phi_l|_1,
$$

where $\lambda$ controls overall regularization magnitude. The problem is in the sparsification which is claimed to be critical to KAN's interpretability. For efficiency, Efficient-KAN [3] instead replaces the L1 regularization on samples with the L1 regularization on the weights, which is more common in neural networks and is compatible with the reformulation:

$$
\mathcal{L}_{\text{total}} = \mathcal{L}_{\text{pred}} + \lambda \sum_{l=1}^{L} \|\mathbf{W}_l\|_1.
$$

$L_2$-**regularization**  $L_2$ regularization promotes smooth optimization landscapes and improves generalization by penalizing large weights. The training objective is formulated as:

$$
\mathcal{L}_{\text{total}} = \mathcal{L}_{\text{pred}} + \lambda \sum_{l=0}^{L-1} \|\mathbf{W}_l\|_2.
$$

---

[3] https://github.com/Blealtan/efficient-kan

Theoretically, $L_2$ regularization interacts with the implicit regularization induced by residual connections, further constraining the optimization path to smoother, more stable solutions. This synergy ensures weight shrinkage toward zero while preserving gradient propagation through skip connections [Zaeemzadeh et al., 2020, Neyshabur, 2017].

**Dropout**    Dropout [Srivastava et al., 2014] is a popular and effective heuristic for preventing large neural networks from overfitting. Indeed, dropout improves the stability bounds generically [Hardt et al., 2016]. From the point of view of stochastic gradient descent, dropout is equivalent to setting a fraction of the gradient weights to zero. In our paper, $L_{1.5}$-regularization also improve the bounds.

$L_{1.5}$**-regularization**    The design of $L_{1.5}$-loss originates from generalization bounds (Theorem 4.5), which regularize $\|\mathbf{W}_i^{\mathrm{T}}\|_{2,2,1}$ in $R_{\mathcal{W}_1^L}$, leading to an improved bias-variance trade-off (Corollary E.1).

## C    Supplementary Definition

### C.1    Norms of Vectors, Matrices and Three-Dimensional Tensors

In this section, we define the norms of vectors (e.g., $\boldsymbol{x_i}$), matrices (e.g., the input matrix $X$) and three-dimensional tensors (e.g., the weight tensor $\mathbf{W}_i$ of KANs). Let $1 \leq p, r \leq \infty$. Given a vector $\alpha = (a_1, a_2 \ldots, a_m)^T \in \mathbb{R}^m$, the $L_p$ norm of $\alpha$ is $\|\alpha\|_p = \left(\sum_i^m a_i^p\right)^{\frac{1}{p}}$. Given a matrix $A = (\alpha_1, \alpha_2, \ldots, \alpha_n) \in \mathbb{R}^{m \times n}$ where the i-th column of $A$ is $\alpha_i \in \mathbb{R}^m$, we use $\|A\|_{p,r} = \|(\|\alpha_1\|_p, \|\alpha_2\|_p, \ldots, \|\alpha_n\|_p)\|_r$ to represent the $(p, r)$ norm of matrix $A$, denoted as $\|A\|_{p,r}$. Especially, the standard $L_1$ induced norm and $L_\infty$ induced norm of matrix $A$ can be represented respectively by $\|A\|_{1,\infty}$ and $\|A\|_{\infty,1}$. Besides, we use $\|A\|_\sigma$ to represent the standard spectral norm (also called $L_2$ induced norm) of $A$. To define norms for three-dimensional tensors similarly to the matrix $(p, r)$ norms, we can generalize the structure systematically. Let $\mathcal{A} \in \mathbb{R}^{m \times n \times k}$ be a three-dimensional tensor. Its elements are denoted by $\mathcal{A}_{ijk}$. The tensor $\mathcal{A}$ can be thought of as a stack of matrices, where the $k$-th matrix slice is $\mathcal{A}_{::k}$ (i.e., fixing the third index of the tensor). Let $1 \leq p, r, q \leq \infty$, we define the $(p, r, q)$ norm of $\mathcal{A}$ as: $\|\mathcal{A}\|_{p,r,q} = \left(\sum_{k=1}^p \|\mathcal{A}_{::k}\|_{p,r}^q\right)^{1/q}$, where $\|\mathcal{A}_{::k}\|_{p,r} = \left\|\left(\|\mathcal{A}_{:1k}\|_p, \|\mathcal{A}_{:2k}\|_p, \ldots, \|\mathcal{A}_{:nk}\|_p\right)\right\|_r$ for each slice $\mathcal{A}_{::k} \in \mathbb{R}^{m \times n}$.

### C.2    Sub-Gaussian Distribution

Our analysis focuses on sub-Gaussian distribution $\mathbb{P}$, so we introduce sub-Gaussian random variables and sub-Gaussian random vectors for preparation. Given a random variable $x$, if there exists $K > 0$ such that the tail of $x$ satisfies $\mathbb{P}\{|x| \geq t\} \leq 2 \exp\left(-t^2/K^2\right), \forall t \geq 0$, then we call $x$ a sub-Gaussian random variable, where the quantity $K^2$ is named the sub-Gaussian variance proxy. The sub-Gaussian norm of $x$, denoted $\|x\|_{\psi_2}$, is defined as $\|x\|_{\psi_2} := \inf\left\{t > 0 : \mathbb{E}\left[\exp\left(x^2/t^2\right) \leq 2\right]\right\}$. It can be deduced by Markov's inequality that there exists $c > 0$ such that $\mathbb{P}\{|x| \geq t\} \leq 2 \exp\left(-ct^2/\|x\|_{\psi_2}^2\right)$, indicating that the tail decays slower as the sub-Gaussian norm becomes larger. A random vector $\boldsymbol{x}$ in $\mathbb{R}^n$ is called *sub-Gaussian* if the one-dimensional marginals $\langle X, x \rangle$ are sub-Gaussian random variables for all $x \in \mathbb{R}^n$. The sub-Gaussian norm of $X$ is defined as $\|\boldsymbol{x}\|_{\psi_2} := \sup_{\boldsymbol{y} \in S^{n-1}} \|\langle \boldsymbol{x}, \boldsymbol{y} \rangle\|_{\psi_2}$. In particular, a random vector with independent bounded coordinates is a sub-Gaussian random vector.

### C.3    Lipschitz Norm

In this section, we discuss Lipschitz properties of various function family.

**Definition C.1** (Lipschitz functions). Let $(X, d_X)$ and $(Y, d_Y)$ be metric spaces. A function $f : X \to Y$ is called *Lipschitz* if there exists $L \in \mathbb{R}$, such that

$$d_Y(f(u), f(v)) \leq L \cdot d_X(u, v), \forall u, v \in X.$$

The infimum of all $L$ in this definition is called the Lipschitz norm of $f$ and is denoted $\|f\|_{\mathrm{Lip}}$.

In other words, Lipschitz functions may not blow up distances between points too much. Lipschitz functions with $\|f\|_{\text{Lip}} \le 1$ are usually called contractions. Specifically, the layer normalization $\tilde{\boldsymbol{x}} = \tanh(\boldsymbol{x})$ is a contraction with $\tilde{x}_i = \frac{e^{x_i} - e^{-x_i}}{e^{x_i} + e^{-x_i}}$.

**Lemma C.1.** *If $\sigma : \mathbb{R}^d \to \mathbb{R}^d$ is $\rho$-Lipschitz along every coordinate, then it is $\rho$-Lipschitz according to $\| \cdot \|_p$ for any $p \ge 1$.*

*Proof.* For any $\boldsymbol{x}, \boldsymbol{x}' \in \mathbb{R}^d$,

$$
\|\sigma(\boldsymbol{x}) - \sigma(\boldsymbol{x}')\|_p = \left( \sum_i |\sigma(\boldsymbol{x})_i - \sigma(\boldsymbol{x}')_i|^p \right)^{1/p} \le \left( \sum_i \rho^p |\boldsymbol{x}_i - \boldsymbol{x}'_i|^p \right)^{1/p} = \rho \|\boldsymbol{x} - \boldsymbol{x}'\|_p
\tag{13}
$$

$\square$

**Lemma C.2.** *Let $\Sigma = \{\sigma_i : [-1, 1] \to \mathbb{R} \mid i \in [G]\}$ be a function family. If $\sigma_i$ is $\rho_i$-Lipschitz, then $M_\Sigma : \mathbb{R}^d \to \mathbb{R}^{d \times G}$ is $\left( \sum_{i=1}^G \rho_i^p \right)^{\frac{1}{p}}$-Lipschitz according to $\| \cdot \|$ for any $p \ge 1$, where the $(i, j)$-th element of the matrix $M_\Sigma(\boldsymbol{x})$ is given by $f_j(x_i)$.*

*Proof.* For any $\boldsymbol{x}, \boldsymbol{x}' \in \mathbb{R}^d$,

$$
\|M_\Sigma(\boldsymbol{x}) - M_\Sigma(\boldsymbol{x}')\|_p = \left( \sum_i \|\sigma_i(\boldsymbol{x}) - \sigma_i(\boldsymbol{x}')\|_p^p \right)^{\frac{1}{p}} \le \left( \sum_i \rho_i^p \|\boldsymbol{x} - \boldsymbol{x}'\|_p^p \right)^{\frac{1}{p}}
$$

$$
= \left( \sum_i \rho_i^p \right)^{\frac{1}{p}} \|\boldsymbol{x} - \boldsymbol{x}'\|_p = \|\rho\|_p \|\boldsymbol{x} - \boldsymbol{x}'\|_p,
$$

where $\rho := (\rho_1, \ldots, \rho_G)^T$.
$\square$

Therefore, we define $\|\Sigma\|_{\text{Lip}} := \|M_\Sigma\|_{\text{Lip}}$ for brevity. We lead to Definition 4.2 when we take $p = 2$.

Now we prove Lemma E.1.

**Lemma E.1** If $\sigma : \mathbb{R} \to \mathbb{R}$ and $l : \mathbb{R} \to \mathbb{R}$ are both Lipschitz continuous functions with Lipschitz norms $\|\sigma\|_{\text{Lip}}$ and $\|l\|_{\text{Lip}}$, respectively, then the composition $l \circ \sigma(x)$ is Lipschitz continuous with Lipschitz norm $\|l \circ \sigma\|_{\text{Lip}} \le \|l\|_{\text{Lip}} \|\sigma\|_{\text{Lip}}$.

*Proof.* By the definition of Lipschitz continuity and Lipschitz norm, for any $x, x' \in \mathbb{R}$,

$$
|l(\sigma(x)) - l(\sigma(x'))| \le \|l\|_{\text{Lip}} |\sigma(x) - \sigma(x')|
$$
$$
\le \|l\|_{\text{Lip}} \|\sigma\|_{\text{Lip}} |x - x'|
$$

Thus, the composition $l \circ \sigma$ satisfies the Lipschitz norm $\|l \circ \sigma\|_{\text{Lip}} \le \|l\|_{\text{Lip}} \|\sigma\|_{\text{Lip}}$.
$\square$

Below, we discuss the Lipschitz property of various basis function families.

- When $\Sigma$ is a set of univariate B-spline basis functions with degree $p$ and knots $\{\xi_i\}_k$, we have $\rho_i \le \frac{2p}{\Delta}$ where $\Delta = \max_i(\xi_{k+p} - \xi_k)$. By default of [Liu et al., 2025], we set $p = 3$ with grid size 5. To ensure the continuity and smoothness of the interpolation, it is usually necessary to extend $p$ points at each end of the original data points. We get the knots $\xi_k$ as

$$
[-2.2, -1.8, -1.4, -1.0, -0.6, -0.2, 0.2, 0.6, 1.0, 1.4, 1.8, 2.2]
$$

with $\Delta = 1.2$ and $\rho_k \le \frac{2p}{\Delta} = 5$. See Figure 2 for the visualization of B-spline basis functions. Finally, the Lipschitz norm of $\Sigma$ can be bounded as $\|\Sigma\|_{\text{Lip}} \le 10\sqrt{2}$.

- When $\Sigma = \{\cos(kx), \sin(kx)\}_k$ is a set of Fourier basis functions[Xu et al., 2024a], we have $\rho_k \le k$ based on Lemma C.3. To ensure the consistency of the number of parameters, we set $k \in [4]$ and deduce that $\|\Sigma\|_{\text{Lip}} \le 2\sqrt{15}$.

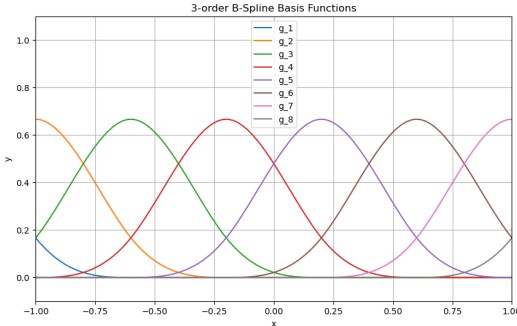

Figure 2: Visualization of the 3-order B-spline basis functions.

**Lemma C.3.** $\cos(kx)$ *and* $\sin(kx)$ *are $k$-Lipschitz continuous.*

*Proof.* We only prove the case of the cosine function; the sine function is similar. For any $x, x' \in \mathbb{R}$,

$$
\begin{aligned}
|\cos(kx) - \cos(ky)| &= \left| -2\sin\left(\frac{kx+ky}{2}\right)\sin\left(\frac{kx-ky}{2}\right) \right| \\
&= 2\left|\sin\left(\frac{kx+ky}{2}\right)\right|\left|\sin\left(\frac{kx-ky}{2}\right)\right| \\
&\leq 2\left|\sin\left(\frac{kx-ky}{2}\right)\right| \\
&\leq 2\left|\frac{kx-ky}{2}\right| \\
&= k|x-y|
\end{aligned}
\tag{14}
$$

where the last inequality is due to the fact that $\left|\sin\left(\frac{kx-ky}{2}\right)\right| \leq \left|\frac{kx-ky}{2}\right|$. $\qquad\square$

- When $\Sigma = \{T_k\}_k$ is a set of Chebyshev polynomial[SS et al., 2024], which is defined as $T_k(x) = \cos(k\arccos(x))$ (please see Figure 3 for the visualization) and can also be expressed using the explicit polynomial form:

$$
\begin{aligned}
T_0(x) &= 1 \\
T_1(x) &= x \\
T_2(x) &= 2x^2 - 1 \\
T_3(x) &= 4x^3 - 3x \\
T_n(x) &= 2xT_{n-1}(x) - T_{n-2}(x) \text{ for } n \geq 2,
\end{aligned}
$$

we can calculate the Lipschitz norm directly due to its polynomial nature. These polynomials are a sequence of orthogonal functions, so only a smaller number of basis functions are needed. We set $|\Sigma| = 4$ like [SS et al., 2024] and in this case, $\|\Sigma\|_{\text{Lip}} \leq \sqrt{14}$.

## C.4 Einsum

To define the product between tensors, we first introduce the definition of matrix inner product.

The inner product of two matrices $A, B \in \mathbb{R}^{m \times n}$ is defined as the sum of the products of their corresponding entries. Mathematically, it is given by:

$$
\langle A, B \rangle = \sum_{i=1}^{m}\sum_{j=1}^{n} A_{ij}B_{ij}.
$$

Alternatively, this can be expressed using the trace of the product of $A$ and $B^T$

$$
\langle A, B \rangle = \text{tr}\left(A^T B\right).
$$

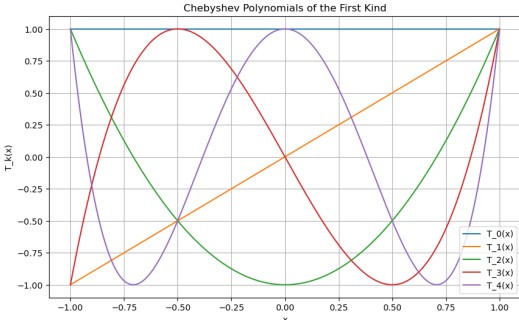

Figure 3: Visualization of the Chebyshev polynomials of the first kind.

Specifically, the inner product induces the Frobenius norm:

$$\|A\|_F = \sqrt{\langle A, A \rangle},$$

with the Cauchy-Schwarz inequality stated as

$$|\langle A, B \rangle| \leq \|A\|_F \cdot \|B\|_F. \tag{15}$$

Below we define the Einstein summation operation(einsum) "$\circ$" between $W \in \mathbb{R}^{m,n,l}$ and $\Sigma \in \mathbb{R}^{n,l}$ as

$$\text{einsum}(\mathbf{W}, \Sigma)_i = (\mathbf{W} \circ \Sigma)_i := \sum_{j=1}^{n} \sum_{k=1}^{l} W_{i,j,k} \Sigma_{j,k}, i = 1, \dots, m.$$

We can also represent $W \circ X \in \mathbb{R}^m$ as $(W \circ \Sigma)_i = \langle W_i, X \rangle$ where $W_i = W_{i::} \in \mathbb{R}^{n,l}$.

## D  Supplementary Experiments

### D.1  Function Approximation

**Datasets**  Feynman [Udrescu et al., 2020] is a symbolic regression and function approximation dataset of physical equations collected from Feynman's textbook. We implement a function approximation task using the Feynman dataset, as suggested by Liu et al. [2025] to investigate whether LipKANs can learn better activation functions compared to KANs.

To demonstrate that LipKANs preserves the powerful expressive capacity of KANs, we first compare the function approximation capabilities of LipKANs with baselines. We utilize the function generation script provided by PyKAN [Liu et al., 2025] and assess model performance on the Feynman dataset. We exclude some equations since the function generation script returns NaN in input and inf in label (such as "Jackson 11.38 (Doppler)"). In Table 3, we present the RMSE of each model on several

Table 3: RMSE Comparison between KANs (the first row of each cell) and LipKANs (the second row of each cell) on Feynman Dataset.

| Feynman Eq. | Original Formula | MLP | KAN | Fourier-KAN | Rational-KAN | RBF-KAN | Cheby-KAN |
|---|---|---|---|---|---|---|---|
| I.6.20a | $\frac{e^{-\frac{x^2}{2}}}{\sqrt{2\pi}}$ | 0.04859 | 0.00047 | 0.02359 | 0.00219 | 0.00082 | 0.04859 |
| | | | **0.00039** | 0.02884 | 0.00385 | 0.00482 | 0.04761 |
| I.9.18 | $\frac{G \cdot m_1 \cdot m_2}{(x_2-x_1)^2+(y_2-y_1)^2+(z_2-z_1)^2}$ | 0.42783 | 0.41967 | 0.54147 | **0.41817** | 0.42203 | 0.41933 |
| | | | 0.41965 | 0.49766 | 0.42004 | 0.41955 | 0.41954 |
| I.24.6 | $\frac{1}{4}m\left(\omega^2 + \omega_0^2\right)x^2$ | 0.04384 | 0.00877 | 0.10813 | 0.02019 | 0.01101 | 0.04750 |
| | | | **0.00871** | 0.07405 | 0.03187 | 0.03415 | 0.04821 |
| I.13.4 | $\frac{1}{2}m\left(v^2 + u^2 + w^2\right)$ | 0.23311 | 0.01264 | 0.27906 | 0.01203 | 0.05021 | 0.26101 |
| | | | **0.01187** | 0.17017 | 0.07661 | 0.09352 | 0.23197 |
| II.35.18 | $\frac{n_0}{\exp\left(\frac{\mu B}{k_B T}\right)+\exp\left(-\frac{\mu B}{k_B T}\right)}$ | 0.13784 | 0.06437 | 0.15890 | 0.04542 | **0.03442** | 0.17830 |
| | | | 0.06220 | 0.14590 | 0.08346 | 0.07837 | 0.17846 |
| III.4.33 | $\frac{\hbar\omega}{\exp\left(\frac{\hbar\omega}{k_B T}\right)-1}$ | 0.27705 | **0.18712** | 0.42472 | 0.21685 | 0.19196 | 0.33203 |
| | | | 0.18774 | 0.29129 | 0.20695 | 0.21895 | 0.35953 |

examples, where the second row of each cell represents the Lip version of the corresponding model. It is clear in Table 3 that the Lip version of KANs not only does not diminish the expressive power

but even outperforms in many functions. Among all the baselines, the original KAN achieves the best fitting performance, while Fourier-KAN exhibits the worst fitting performance. Rational-KAN and RBF-KAN perform closely in RMSE. MLP is not comparable to the KAN-based architecture in the experiment, reflecting the strong function approximation ability of KAN-based methods.

## D.2  Generalization Gap

In this section, to provide a more comprehensive analysis of generalization performance, we follow the prior works [Hoffer et al., 2017, Fu et al., 2023] and quantify generalization gaps using the following three metrics:

- Final Generalization Gap: Difference between training and validation accuracy at convergence.
- Average Generalization Gap: Mean gap across the entire training trajectory.
- Overfitting Ratio: Final gap normalized by final training accuracy.

The results are summarized below.

Table 4: Analysis of generalization performance using four metrics across datasets: LipKANs achieves consistent advantages in generalization gaps and overfitting ratio.

| Dataset | Model | Final Gap (%) | Avg. Gap (%) | Overfit Ratio (%) | Val. Acc (%) |
|---|---|---|---|---|---|
| CIFAR-10 | KANs ($L_1$-Regularization) | 6.04 | 5.64 | 13.13 | 46.01 |
| | KANs + Lip Layers | 4.45 | 4.92 | 9.47 | 46.99 |
| | KANs + $L_{1.5}$-Regularization | 4.99 | 5.28 | 10.11 | 49.37 |
| | LipKANs | **3.16** | **4.89** | **6.26** | **50.40** |
| CIFAR-100 | KANs ($L_1$-Regularization) | 6.55 | 5.13 | 44.35 | 14.77 |
| | KANs + Lip Layers | 5.84 | 5.10 | 38.98 | 14.98 |
| | KANs + $L_{1.5}$-Regularization | 3.76 | 5.09 | 19.89 | 19.93 |
| | LipKANs | **2.32** | **5.03** | **11.31** | **20.51** |
| MNIST | KANs ($L_1$-Regularization) | 5.52 | 4.92 | 5.77 | 95.62 |
| | KANs + Lip Layers | 3.17 | 4.77 | 3.30 | 96.06 |
| | KANs + $L_{1.5}$-Regularization | 5.13 | 4.81 | 5.27 | 97.26 |
| | LipKANs | **2.55** | **4.70** | **2.60** | **97.71** |
| STL-10 | KANs ($L_1$-Regularization) | 6.93 | 5.27 | 26.72 | 25.94 |
| | KANs + Lip Layers | 4.14 | 5.10 | 15.54 | 26.64 |
| | KANs + $L_{1.5}$-Regularization | 3.95 | 5.06 | 10.47 | 37.73 |
| | LipKANs | **3.31** | **4.98** | **8.53** | **38.77** |

Across all datasets, the LipKANs consistently achieve the lowest overfitting ratio and final generalization gap, suggesting stronger generalization. Interestingly, this advantage does not always manifest as significantly higher validation accuracy—some models achieve similar validation accuracy, yet differ notably in their generalization gaps. This discrepancy reflects a more favorable bias–variance trade-off. Specifically, models with higher variance may fit the training data well yet generalize poorly, resulting in larger generalization gaps despite acceptable validation accuracy. In contrast, regularization methods such as those employed in LipKANs may introduce slight bias but effectively reduce variance, leading to more reliable generalization behavior.

## D.3  Computational Cost

We would like to clarify that both the Lip layer and the $L_{1.5}$-regularization are lightweight mechanisms that introduce minimal computational burden. The Lip layer imposes Lipschitz norm constraints to control the smoothness of basis functions, without modifying the network architecture or introducing auxiliary parameters. The $L_{1.5}$-regularization is a direct substitute for the commonly used $L_1$ penalty—it operates on the same weight tensors already present in training and does not incur additional memory overhead.

Empirically, in Table 5 we quantify efficiency using two widely adopted metrics: training time (in seconds) and peak GPU memory usage (in MB). Our measurements show that both the Lipschitz

Table 5: Comprehensive ablation study on computational cost (training time and memory) across datasets.

| Model | CIFAR-10 | | CIFAR-100 | | MNIST | | STL-10 | |
|---|---|---|---|---|---|---|---|---|
| | Time (s) | Memory (MB) | Time (s) | Memory (MB) | Time (s) | Memory (MB) | Time (s) | Memory (MB) |
| KANs ($L_1$-Regularization) | 263.96 | 53.52 | 265.45 | 54.61 | 227.72 | 26.44 | 221.57 | 356.28 |
| KANs + Lip Layers | 268.88 | 53.52 | 259.99 | 54.61 | 226.14 | 26.44 | 227.49 | 356.40 |
| KANs ($L_{1.5}$-Regularization) | 262.73 | 53.52 | 283.43 | 54.63 | 227.00 | 26.57 | 221.62 | 356.63 |
| LipKANs | 264.89 | 53.52 | 263.72 | 54.63 | 229.04 | 26.57 | 226.76 | 356.65 |

layers and the $L_{1.5}$-regularization incur only moderate overhead relative to baseline KANs, while delivering notable improvements in generalization performance.

Table 6: Model performance across datasets with different grid sizes and spline orders.

| Dataset | Grid Size | Spline Order | #Params | Final Gen Gap (%) | Test Acc (%) | Overfit Ratio (%) | Train Time (s) | Peak GPU Mem |
|---|---|---|---|---|---|---|---|---|
| | 3 | 2 | 1,387,008 | 1.42 | 50.42 | 2.81 | 196.96 | 39.57 |
| | 3 | 3 | 1,584,256 | 0.79 | 49.60 | 1.59 | 201.86 | 42.17 |
| | 3 | 4 | 1,781,504 | 0.59 | 49.51 | 1.19 | 212.61 | 45.08 |
| | 5 | 2 | 1,781,504 | 4.00 | 52.39 | 7.64 | 201.16 | 45.05 |
| CIFAR-10 | 5 | 3 | 1,978,752 | 3.06 | 52.33 | 5.85 | 210.33 | 48.03 |
| | 5 | 4 | 2,176,000 | 2.30 | 52.15 | 4.41 | 221.63 | 50.50 |
| | 7 | 2 | 2,176,000 | 6.94 | 53.19 | 13.05 | 203.14 | 50.47 |
| | 7 | 3 | 2,373,248 | 5.78 | 52.94 | 10.92 | 215.97 | 53.51 |
| | 7 | 4 | 2,570,496 | 4.50 | 53.23 | 8.45 | 228.08 | 56.85 |
| | 3 | 2 | 1,427,328 | 1.47 | 22.00 | 6.68 | 202.60 | 40.22 |
| | 3 | 3 | 1,630,336 | 1.65 | 21.08 | 7.82 | 198.90 | 42.90 |
| | 3 | 4 | 1,833,344 | 1.52 | 21.00 | 7.24 | 211.00 | 45.90 |
| | 5 | 2 | 1,833,344 | 3.09 | 23.29 | 13.27 | 197.98 | 45.87 |
| CIFAR-100 | 5 | 3 | 2,036,352 | 2.57 | 23.57 | 10.91 | 204.95 | 48.93 |
| | 5 | 4 | 2,239,360 | 2.48 | 22.59 | 10.98 | 220.06 | 51.49 |
| | 7 | 2 | 2,239,360 | 4.66 | 24.63 | 18.93 | 201.39 | 51.47 |
| | 7 | 3 | 2,442,368 | 4.05 | 24.12 | 16.78 | 214.83 | 54.59 |
| | 7 | 4 | 2,645,376 | 3.92 | 24.04 | 16.31 | 227.14 | 58.02 |
| | 3 | 2 | 357,408 | 4.12 | 97.28 | 4.24 | 25.06 | 50.66 |
| | 3 | 3 | 408,224 | 3.49 | 97.59 | 3.58 | 26.35 | 52.84 |
| | 3 | 4 | 459,040 | 3.14 | 97.54 | 3.22 | 29.00 | 54.18 |
| | 5 | 2 | 459,040 | 3.56 | 97.33 | 3.66 | 31.87 | 54.18 |
| MNIST | 5 | 3 | 509,856 | 3.31 | 97.40 | 3.40 | 33.82 | 55.52 |
| | 5 | 4 | 560,672 | 2.89 | 97.45 | 2.97 | 35.92 | 56.86 |
| | 7 | 2 | 560,672 | 3.36 | 97.41 | 3.45 | 38.63 | 56.86 |
| | 7 | 3 | 611,488 | 2.88 | 97.47 | 2.96 | 41.45 | 58.20 |
| | 7 | 4 | 662,304 | 2.91 | 97.47 | 2.99 | 43.99 | 59.54 |
| | 3 | 2 | 1,387,008 | 0.78 | 28.54 | 2.73 | 98.73 | 39.58 |
| | 3 | 3 | 1,584,256 | 1.20 | 28.10 | 4.27 | 95.53 | 42.73 |
| | 3 | 4 | 1,781,504 | 0.98 | 27.50 | 3.56 | 96.58 | 46.39 |
| | 5 | 2 | 1,781,504 | 1.45 | 32.41 | 4.47 | 93.69 | 46.37 |
| STL-10 | 5 | 3 | 1,978,752 | 1.62 | 31.20 | 5.19 | 95.50 | 48.03 |
| | 5 | 4 | 2,176,000 | 1.01 | 31.25 | 3.23 | 97.00 | 51.81 |
| | 7 | 2 | 2,176,000 | 1.68 | 34.40 | 4.88 | 93.14 | 51.79 |
| | 7 | 3 | 2,373,248 | 1.95 | 33.91 | 5.75 | 97.27 | 54.07 |
| | 7 | 4 | 2,570,496 | 1.73 | 33.73 | 5.13 | 98.40 | 57.10 |

## D.4 Activation Family Size

The activation family size $G$ reflects the cardinality of the basis function family, treated as a constant that captures overall family size, while its internal structure (e.g., smoothness) remains tunable. For instance, KANs can employ different basis families—B-splines (original KAN), Fourier (Fourier-KAN), and rational functions (Rational-KAN)—each with adjustable basis affect generalization without altering $G$.

From a practical perspective, we conducted sensitivity analyses on B-spline basis functions by varying degree and grid size in Table 6, which together determine the shape and cardinality of the activation family. The key findings are:

1. Internal structure matters beyond size: Even when $G =$ Grid size + Spline order is fixed, generalization can vary significantly. On MNIST, both (3,4) and (5,2) yield the same $G$, yet the Final Gen Gap differs by over 7×—highlighting that basis structure, not just size, plays a crucial role.

2. Opposing effects of grid size vs. spline order: With fixed grid size, increasing spline order improves generalization; conversely, with fixed spline order, increasing grid size degrades

generalization. This aligns with our view that smoother basis functions (higher order) help regulate model complexity.

3. Efficiency trade-offs: Larger $G$ increases computational cost, with spline order contributing more to time and memory overhead than grid size.

These results reinforce our theoretical claims and demonstrate the framework's robustness across different architectural settings.

# E  Proof of Main Results

## E.1  Proof of Lemma 4.2

As outlined in the text, constructing a whole-network cover through induction on layers requires minimal assumptions about the norms imposed on the weight matrices. In this subsection, we delve into a more general analysis of this approach. The structure of the networks is the same as before; namely, given tensors $\mathcal{W} = (\mathbf{W}_1, \ldots, \mathbf{W}_L)$, define the mapping $F_{\mathcal{W}}$ as $F_{\mathcal{W}}(\boldsymbol{x}) = \mathrm{KAN}_{\mathcal{W}}(\boldsymbol{x})$ in Eq. (1). More generally for $i \leq L$ define $\mathcal{W}_1^i := (\mathbf{W}_1, \ldots, \mathbf{W}_i)$ and

$$F_{\mathcal{W}_1^i}(\boldsymbol{x}) = (\mathbf{W}_i \circ \Sigma \circ \cdots \circ \mathbf{W}_1 \circ \Sigma)\boldsymbol{x}$$

with the convention $F_\emptyset(\boldsymbol{x}) = \boldsymbol{x}$. For brevity, we write $F_{\mathcal{W}_1^i}(\boldsymbol{x})$ as $F_{\mathcal{W}_1^i}$ when data $\boldsymbol{x}$ is fixed.

**Lemma 4.2** Let $(\epsilon_1, \ldots, \epsilon_L)$ be given. Under Assumption 1 and 2, each $\sigma_k$ is Lipschitz continuous with $\rho_k := \|\sigma_k\|_{\mathrm{Lip}}$. Let $\rho := (\rho_1, \ldots, \rho_G)$. Suppose the tensors $\mathcal{W}_1^L = (\mathbf{W}_1, \ldots, \mathbf{W}_L)$ lie within $\mathcal{B}_1 \times \cdots \times \mathcal{B}_L$ where $\mathcal{B}_i$ are arbitrary classes with the property that each $\mathbf{W}_i \in \mathcal{B}_i$ has $\|\mathbf{W}_i\|_\sigma \leq c_i$. Then, letting $\tau_1 = \epsilon_1$ and $\tau_l = \sum_{i=1}^l (\prod_{j=i+1}^l c_j \|\rho\|_2) \epsilon_i$, the neural net images $\mathcal{H}_X := \left\{ F_{\mathcal{W}_1^L}(X) : \mathcal{W}_1^L \in \mathcal{B}_1 \times \cdots \times \mathcal{B}_L \right\}$ have a covering number bound:

$$\mathcal{N}\left(\mathcal{H}_X, \tau_L, \|\cdot\|_2\right) \leq \prod_{i=1}^L \sup_{\substack{(\mathbf{W}_1, \ldots, \mathbf{W}_{i-1}) \\ \forall j < i \cdot \mathbf{W}_j \in \mathcal{B}_j}} \mathcal{N}\left(\left\{\mathbf{W}_i \circ \Sigma\left(F_{\mathcal{W}_1^{i-1}}(X)\right) : \mathbf{W}_i \in \mathcal{B}_i\right\}, \epsilon_i, \|\cdot\|_2\right).$$

*Proof.* Inductively construct covers $\mathcal{F}_1, \ldots, \mathcal{F}_L$ of $\mathcal{W}_2, \ldots, \mathcal{W}_{L+1}$ as follows,

- Choose an proper $\epsilon_1$-cover $F_{\mathcal{W}_1^1}$ of $\{F_{\mathcal{W}_1^1}, \mathbf{W}_1 \in \mathcal{B}_1\}$, thus

$$|\mathcal{F}_1| = \mathcal{N}(\{F_{\mathcal{W}_1^1}, \mathbf{W}_1 \in \mathcal{B}_1\}, \epsilon_1, \|\cdot\|_2) =: N_1$$

- For every element $F \in \mathcal{F}_1$, construct an $\epsilon_{i+1}$-cover $\mathcal{G}_{i+1}(F)$ of

$$\{\mathbf{W}_{i+1} \circ \Sigma(F) : \mathbf{W}_{i+1} \in \mathcal{B}_{i+1}\}.$$

The covers are proper, so $F = F_{\mathcal{W}_1^{i-1}}$ for some $(\mathbf{W}_1, \ldots, \mathbf{W}_i) \in \mathcal{B}_1 \times \cdots \times \mathcal{B}_i$. It follows that

$$|\mathcal{G}_{i+1}(F)| \leq \sup_{\substack{(\mathbf{W}_1, \ldots, \mathbf{W}_i) \\ \forall j \leq i, \mathbf{W}_j \in \mathcal{B}_j}} \mathcal{N}\left(\left\{\mathbf{W}_{i+1} F_{\mathcal{W}_1^i}(Z) : \mathbf{W}_{i+1} \in \mathcal{B}_{i+1}\right\}, \epsilon_{i+1}, \|\cdot\|_{i+2}\right) =: N_{i+1}$$

Construct the cover

$$\mathcal{F}_{i+1} := \cup_{F \in \mathcal{F}_i} \mathcal{G}_{i+1}(F), \tag{16}$$

with cardinality

$$|\mathcal{F}_{i+1}| \leq |\mathcal{F}_i| \cdot N_{i+1} \leq \prod_{l=1}^{i+1} N_l.$$

Below, we show that $\mathcal{F}_{i+1}$ is an $\tau_{i+1}$-cover of $\{F_{\mathcal{W}_1^{i+1}} : (\mathbf{W}_1 \times \cdots \times \mathbf{W}_{i+1}) \in \mathcal{B}_1 \times \cdots \times \mathcal{B}_{i+1}\}$. For any $F_{\mathcal{W}_1^{i+1}} = (\mathbf{W}_{i+1} \circ \Sigma \circ \cdots \circ \mathbf{W}_1 \circ \Sigma)\boldsymbol{x}$, by construction, we can find $\hat{F}_1 \in \mathcal{F}_1$ such that

$$\|\hat{F}_1 - F_{\mathcal{W}_1^1}\| \leq \epsilon_1 = \tau_1.$$

Now suppose we can find $\hat{F}_i \in \mathcal{F}_i$ such that

$$\|\hat{F}_i - F_{\mathcal{W}_1^i}\| \le \tau_i$$

By the construction of $\mathcal{F}_{i+1}$ in Eq. (16), we can find $\hat{F}_{i+1} \in \mathcal{F}_{i+1}$ such that

$$\|\hat{F}_{i+1} - \mathbf{W}_{i+1} \circ \Sigma(\hat{F}_i)\|_2 \le \epsilon_{i+1}$$

Then we use the triangle inequality of norm $\|\cdot\|_2$, we have

$$\begin{aligned}
\|\hat{F}_{i+1} - F_{\mathcal{W}_1^{i+1}}\|_2 =& \|\hat{F}_{i+1} - \mathbf{W}_{i+1} \circ \Sigma(\hat{F}_i) + \mathbf{W}_{i+1} \circ \Sigma(\hat{F}_i) - F_{\mathcal{W}_1^i}\|_2 \\
\le& \|\hat{F}_{i+1} - \mathbf{W}_{i+1} \circ \Sigma(\hat{F}_i)\|_2 + \|\mathbf{W}_{i+1} \circ \Sigma(\hat{F}_i) - F_{\mathcal{W}_1^{i+1}}\|_2 \\
\le& \epsilon_{i+1} + \|\mathbf{W}_{i+1}\|_\sigma \|\Sigma(\hat{F}_i - F_{\mathcal{W}_1^i})\|_2 \\
\le& \epsilon_{i+1} + c_{i+1} \|\rho\|_2 \tau_i = \tau_{i+1}
\end{aligned}$$

where the second $\le$ is due to Cauchy-Schwarz inequality in Eq. (15). $\qquad\square$

## E.2 Proof of Lemma 4.3

The proof relies upon the Lemma A.3, which is stated in terms of sparsifying convex hulls, and in its use here is inspired by covering number bounds for linear predictors.

**Lemma 4.3** Let conjugate exponents $(p, q)$ and $(u, v)$ be given with $p \le 2$ as well as positive reals $(a, b, \epsilon)$ and positive integer $m$. Let data $X$ be given with $\|\Sigma(X)\|_p \le b$. Then

$$\ln \mathcal{N}\left(\left\{\mathbf{W} \circ \Sigma(X) : \mathbf{W} \in \mathbb{R}^{d' \times d \times G}, \|\mathbf{W}\|_{q,q,v} \le a\right\}, \epsilon, \|\cdot\|_2\right) \le \left\lceil \frac{a^2 b^2 d'^{\frac{2}{u}}}{\epsilon^2} \right\rceil \ln(2dd'G),$$

where $\lceil x \rceil$ returns the smallest integer greater than or equal to $x$.

*Proof.* Let data $X \in \mathbb{R}^{n \times d}$ be given, with weight tensor $\mathbf{W} \in \mathbb{R}^{d \times G \times d'}$. For brevity, we write tensor $\hat{\Sigma} := \Sigma(X) \in \mathbb{R}^{n \times d \times G}$ with $\hat{\Sigma}_{ijk} = [g_k(\boldsymbol{x}_i)]_j$ and $\mathbf{W} \circ \Sigma(\boldsymbol{x}) \in \mathbb{R}^{n \times d'}$. Set $k := \left\lceil \frac{a^2 b^2 d'^{\frac{2}{u}}}{\epsilon^2} \right\rceil$.

We obtain tensor $\bar{\Sigma} \in \mathbb{R}^{n \times d \times G}$ by rescaling the first dimension of tensor $\hat{\Sigma}$ to have unit $p$-norm: $\bar{\Sigma}_{:jk} := \frac{\hat{\Sigma}_{:jk}}{\|\hat{\Sigma}_{:jk}\|_p}$.

Define $\alpha \in \mathbb{R}^{d \times G \times d'}$ to be a rescaling tensor such that for each $i \in [d']$,

$$\alpha_{::i} = \begin{pmatrix}
\|\Sigma_{:1,1}\|_p & \|\Sigma_{:1,2}\|_p & \cdots & \|\Sigma_{:1,G}\|_p \\
\|\Sigma_{:2,1}\|_p & \ddots & \cdots & \vdots \\
\vdots & \vdots & \ddots & \vdots \\
\|\Sigma_{:d,1}\|_p & \cdots & \cdots & \|\Sigma_{:d,G}\|_p
\end{pmatrix}$$

which satisfies $\mathbf{W} \circ \tilde{\Sigma} = (\alpha \odot \mathbf{W}) \circ \bar{\Sigma}$ where $\odot$ denotes element-wise product. Note that

$$\begin{aligned}
\|\alpha\|_{p,p,u} &= \left(\sum_{i=1}^{d'} \|\alpha_{::i}\|_{p,p}^u\right)^{\frac{1}{u}} \\
&= (d')^{\frac{1}{u}} \|\alpha_{::i}\|_{p,p} \\
&= (d')^{\frac{1}{u}} \left(\sum_j^d \sum_k^G \|\Sigma_{:jk}\|_p^p\right)^{\frac{1}{p}} \\
&= (d')^{\frac{1}{u}} \left(\sum_i^n \sum_j^d \sum_k^G \|\Sigma_{ijk}\|_p^p\right)^{\frac{1}{p}} \\
&= (d')^{\frac{1}{u}} \|\tilde{\Sigma}\|_p
\end{aligned}$$

Define $\bar{\mathbf{W}} = \alpha \odot \mathbf{W}$, whereby using conjugacy of $\|\cdot\|_{p,r,u}$ and $\|\cdot\|_{q,s,v}$ gives

$$\|\bar{\mathbf{W}}\|_1 \leq \langle \alpha, |\mathbf{W}| \rangle \leq \|\alpha\|_{p,p,u} \|\mathbf{W}\|_{q,q,v} \leq (d')^{\frac{1}{u}} \|\Sigma\|_p a =: \bar{a}$$

Now we have

$$
\begin{aligned}
\mathbf{W} \circ \tilde{\Sigma} = \bar{\mathbf{W}} \circ \bar{\Sigma} &= \left( \sum_i^d \sum_j^G \sum_k^{d'} \bar{\mathbf{W}}_{ijk} E_{ijk} \right) \circ \bar{\Sigma} \\
&= \|\bar{\mathbf{W}}\|_1 \left( \sum_i^d \sum_j^G \sum_k^{d'} \frac{\bar{\mathbf{W}}_{ijk}}{\|\bar{\mathbf{W}}\|_1} E_{ijk} \circ \bar{\Sigma} \right)
\end{aligned}
\tag{17}
$$

where $E_{ijk} \in \mathbb{R}^{d \times G \times d'}$ is a tensor where the element at position $(i, j, k)$ is 1, and all other elements are 0.

To derive the desired cover, we define

$$\{V_1, \ldots, V_N\} = \left\{ g E_{ijk} \circ \bar{\Sigma} \mid g \in \{-1, +1\}, i \in [d], j \in [G], k \in [d'] \right\},$$

and construct a cover

$$\mathcal{C} := \left\{ \frac{\bar{a}}{k} \sum_{i=1}^N k_i V_i : k_i \geq 0, \sum_{i=1}^N k_i = k \right\} = \left\{ \frac{\bar{a}}{k} \sum_{j=1}^k V_{i_j} : (i_1, \ldots, i_k) \in [N]^k \right\},$$

where $k_i$'s are integers and $N := 2dd'G$. By construction, we have $|\mathcal{C}| \leq N^k$, and

$$\max_i \|V_i\|_2 \leq \max_{j,k} \frac{\|\Sigma_{:jk}\|_p}{\|\Sigma_{:jk}\|_2} \leq 1,$$

where the last inequality is due to $p \leq 2$.

To use Lemma A.3, We can view $V_i$'s as elements in the Hilbert space $\mathbb{R}^{n \times d'}$ equipped with the inner product $\langle A, \widetilde{A} \rangle = \text{trace}\left( A^\top \widetilde{A} \right)$ defined in Appendix C.4 and norm $\|A\|_2$. Following Eq. (17), $\mathbf{W} \circ \tilde{\Sigma}$ lies in the convex hull of $\{V_1, \ldots, V_N\}$, i.e.

$$\mathbf{W} \circ \tilde{\Sigma} = \bar{\mathbf{W}} \circ \bar{\Sigma} \in \bar{a} \cdot \text{conv}\{V_1, \ldots, V_N\}$$

Combining the preceding constructions with Lemma A.3, there exist nonnegative integers $(k_1, \ldots, k_N)$ with $\sum_i k_i = k$ such that

$$\|\mathbf{W} \circ \tilde{\Sigma} - \frac{\bar{a}}{k} \sum_{i=1}^N k_i V_i\|_2^2 = \|\bar{\mathbf{W}} \circ \bar{\Sigma} - \frac{\bar{a}}{k} \sum_{i=1}^N k_i V_i\|_2^2 \leq \frac{\bar{a}^2}{k} \max_i \|V_i\|_2 \leq \frac{a^2 d'^{\frac{2}{u}} \|\Sigma\|_p^2}{k} \leq \epsilon^2.$$

The desired cover element is thus $\frac{\bar{a}}{k} \sum_i k_i V_i \in \mathcal{C}$ and the result follows. $\qquad\square$

### E.3  Proof of Theorem 4.4

The whole-network covering bound now follows by the general norm covering number in Lemma 4.2, and the matrix covering lemma in Lemma 4.3.

**Theorem 4.4** Under Assumption 2, we can bound

$$\log \mathcal{N}\left(\mathcal{H}_X, \epsilon, \|\cdot\|_2\right) \leq \frac{R^2_{\mathcal{W}_1^L} \log(2\tilde{d}^2 G)}{\epsilon^2},$$

where $\tilde{d} := \max_i d_i$.

*Proof.* Given tensors $\mathcal{W}_1^{i-1} := (\mathbf{W}_1, \ldots, \mathbf{W}_{i-1})$, we define

$$\boldsymbol{X}^{l-1} := F_{\mathcal{W}_1^{i-1}}(\boldsymbol{X}) = (\mathbf{W}_{i-1} \circ \Sigma \circ \cdots \circ \mathbf{W}_1 \circ \Sigma)\boldsymbol{X}$$

Set $p = q = 2, u = \infty, v = 1$ in Lemma 4.3, define $\mathcal{B}_i = \{\mathbf{W} \in \mathbb{R}^{d_i \times d_{i-1} \times G} : \|\mathbf{W}\|_\sigma \leq c_i, \|\mathbf{W}\|_{2,2,1} \leq a_i\}$, then we can decompose the whole covering number based on Lemma 4.2.

$$\log \mathcal{N}\left(\mathcal{H}_{\boldsymbol{x}}, \tau_L, \|\cdot\|_2\right)$$

$$\leq \sum_{i=1}^{L} \sup_{\substack{(\mathbf{W}_1,\ldots,\mathbf{W}_{i-1}) \\ \forall j < i \cdot \mathbf{W}_j \in \mathcal{B}_j}} \log \mathcal{N}\left(\left\{\mathbf{W}_i \circ \left(\Sigma F_{\mathcal{W}_1^{i-1}}(\boldsymbol{x})\right) : \mathbf{W}_i \in \mathcal{B}_i\right\}, \epsilon_i, \|\cdot\|_2\right)$$

$$= \sum_{i=1}^{L} \sup_{\substack{(\mathbf{W}_1,\ldots,\mathbf{W}_{i-1}) \\ \forall j < i \cdot \mathbf{W}_j \in \mathcal{B}_j}} \log \mathcal{N}\left(\left\{\mathbf{W}_i \circ \left(\Sigma F_{\mathcal{W}_1^{i-1}}(\boldsymbol{x})\right) - \mathbf{W}_i \circ \Sigma(\mathbf{0}) : \mathbf{W}_i \in \mathcal{B}_i\right\}, \epsilon_i, \|\cdot\|_2\right) \tag{18}$$

$$= \sum_{i=1}^{L} \sup_{\substack{(\mathbf{W}_1,\ldots,\mathbf{W}_{i-1}) \\ \forall j < i \cdot \mathbf{W}_j \in \mathcal{B}_j}} \log \mathcal{N}\left(\left\{\mathbf{W}_i \circ (\Sigma(\boldsymbol{x}^{l-1}) - \Sigma(\mathbf{0})) : \mathbf{W}_i \in \mathcal{B}_i\right\}, \epsilon_i, \|\cdot\|_2\right)$$

$$\leq \sum_{i=1}^{L} \frac{a_i^2 \|\Sigma(\boldsymbol{x}^{l-1}) - \Sigma(\mathbf{0})\|_2^2}{\epsilon_i^2} \log(2 d_i d_{i-1} G)$$

Below, we use the Lipschitz property of $\Sigma$ to bound the term $\|\Sigma(\boldsymbol{x}^{l-1}) - \Sigma(\mathbf{0})\|_2$ as follows:

$$\|\Sigma(\boldsymbol{x}^{l-1}) - \Sigma(\mathbf{0})\|_2 \leq \|\Sigma\|_{\text{Lip}} \|\boldsymbol{x}^{l-1}\|_2$$
$$\leq \|\Sigma\|_{\text{Lip}} \|\mathbf{W}_{l-1} \circ \Sigma(\boldsymbol{x}^{l-2})\|_2$$
$$\leq \|\Sigma\|_{\text{Lip}} \|\mathbf{W}_{l-1}\|_\sigma \|\Sigma(\boldsymbol{x}^{l-2})\|_2$$
$$\leq \|\Sigma\|_{\text{Lip}} \|\mathbf{W}_{l-1}\|_\sigma \|\Sigma(\boldsymbol{x}^{l-2}) - \Sigma(\mathbf{0}) + \Sigma(\mathbf{0})\|_2$$
$$\leq \|\Sigma\|_{\text{Lip}} \|\mathbf{W}_{l-1}\|_\sigma \left(\|\Sigma(\boldsymbol{x}^{l-2}) - \Sigma(\mathbf{0})\|_2 + \|\Sigma(\mathbf{0})\|_2\right)$$
$$\leq \|\Sigma\|_{\text{Lip}} \|\mathbf{W}_{l-1}\|_\sigma \left(\|\Sigma\|_{\text{Lip}} \|\boldsymbol{x}^{l-2}\|_2 + \|\Sigma\|_\infty\right)$$
$$\leq \left(\|\Sigma\|_{\text{Lip}}^l \|\mathbf{X}\|_2 \prod_{i=1}^{l-1} \|\mathbf{W}_i\|_\sigma\right) + \left(\sum_{j=1}^{l-1} \|\Sigma\|_{\text{Lip}}^j \|\Sigma\|_\infty \left(\prod_{i=l-j}^{l-1} \|\mathbf{W}_i\|_\sigma\right)\right)$$
$$\leq \left(C^l D \prod_{i=1}^{l-1} c_i\right) + \left(\sum_{j=1}^{l-1} C^j E \left(\prod_{i=l-j}^{l-1} c_i\right)\right)$$

where $\|\Sigma\|_\infty := \sqrt{\sum_k^G \|g_k\|_\infty^2}$ represents the $\infty$ norm of function family $\Sigma$. Now we suppose $\|\mathbf{W}_i\|_\sigma \leq \sigma_i$ and $\|\Sigma\|_\infty \leq E$, then we have

$$\|\Sigma(\boldsymbol{x}^{l-1}) - \Sigma(\mathbf{0})\|_2 \leq \left(C^l D \prod_{i=1}^{l-1} c_i\right) + \left(\sum_{j=1}^{l-1} C^j E \left(\prod_{i=l-j}^{l-1} c_i\right)\right) \tag{19}$$

Plugging Eq. (19) into Eq. (18), we obtain that

$$\log \mathcal{N}\left(\mathcal{H}_{\boldsymbol{x}}, \tau_L, \|\cdot\|_2\right) \leq \sum_{i=1}^{L} \frac{a_i^2 \|\Sigma(\boldsymbol{x}^{l-1}) - \Sigma(\mathbf{0})\|_2^2}{\epsilon_i^2} \log(2 d_i d_{i-1} G)$$

$$\leq \sum_{i=1}^{L} \frac{a_i^2 \left(C^l D \prod_{j=1}^{i-1} c_j + \sum_{j=1}^{i-1} C^j E \left(\prod_{k=i-j}^{i-1} c_k\right)\right)^2}{\epsilon_i^2} \log(2 \bar{d}^2 G) \tag{20}$$

Define

$$\alpha_i = a_i^{\frac{2}{3}} \left(\prod_{j=i+1}^{L} C c_j\right)^{\frac{2}{3}} \left(C^l D \prod_{j=1}^{i-1} c_j + \sum_{j=1}^{i-1} C^j E \left(\prod_{k=i-j}^{i-1} c_k\right)\right)^{\frac{2}{3}}$$

and $\tilde{\alpha} = \sum_{i=1}^{L} \alpha_i$. For any $\epsilon > 0$, set

$$\epsilon_i = \frac{\alpha_i \epsilon}{\tilde{\alpha} \prod_{j=i+1}^{L} C c_j}$$

Then we have $s_L = \sum_{i=1}^{L} \left( \prod_{j=i+1}^{L} C c_j \right) \epsilon_i = \epsilon$ and hence

$$\log \mathcal{N} \left( \mathcal{H}_{\boldsymbol{x}}, \epsilon, \|\cdot\|_2 \right) \leq \frac{\tilde{\alpha}^3 \log(2\tilde{d}^2 G)}{\epsilon^2}$$

$\square$

## E.4 Proof of Theorem 4.5

We prove our main result by integrating Theorem 4.4 with standard properties of Rademacher complexity in Section A.

**Theorem 4.5** (Generalization Bounds for KANs) Under Assumption 2 and 3, let fixed Lipschitz activations $\Sigma = \{\sigma_k \mid k \in [G]\}$ and weight tensors $\mathcal{W}_1^L = \{\mathbf{W}_1, \mathbf{W}_2, \ldots, \mathbf{W}_L\}$ be given. Then for $(\boldsymbol{x}, y), (\boldsymbol{x}_1, y_1), \ldots, (\boldsymbol{x}_n, y_n)$ drawn iid from distribution $\mathbb{P}$, we have with probability greater than $1 - \epsilon$,

$$R(F_{\mathcal{W}_1^L}) - \hat{R}(F_{\mathcal{W}_1^L}) \leq \frac{144\sqrt{\zeta}\{\log(nM/(3\sqrt{\zeta})) \vee 1\}}{n} + \sqrt{\frac{4M^2 \log(2/\epsilon)}{n}} + \frac{32M \log(2/\epsilon)}{3n}$$

$$\leq \mathcal{O} \left( \frac{\|X\|_2 R_{\mathcal{W}_1^L} \max_i |\rho(y_i)|}{n} \log(\tilde{d}^2 G) + \sqrt{\frac{1/\epsilon}{n}} \right),$$

where $\zeta = R_{\mathcal{W}_1^L}^2 \|X\|_2^2 \log(2\tilde{d}^2 G) \max_i \rho^2(y_i)$.

*Proof.* Define the loss class $\mathcal{L}(S) := \{(\mathcal{L}(f(\boldsymbol{x}_1), y_1), \ldots, \mathcal{L}(f(\boldsymbol{x}_n), y_n)) \mid f \in \mathcal{H}\}$. By theorem 4.4, we have

$$\log \mathcal{N} \left( \mathcal{L}(S), \epsilon, \|\cdot\|_2 \right) \leq \frac{\tilde{\alpha}^3 \log(2\tilde{d}^2 G) \max_i \rho^2(y_i)}{\epsilon^2} = \frac{\zeta}{\epsilon^2}$$

Lemma A.2 implies that

$$\begin{aligned}
\mathcal{R}\left( \mathcal{L}(S) \right) &\leq \inf_{a > 0} \left( \frac{4a}{\sqrt{n}} + \frac{12}{n} \int_a^{\sqrt{n}M} \sqrt{\frac{\zeta}{\epsilon^2}} d\epsilon \right) \\
&= \inf_{a > 0} \left( \frac{4a}{\sqrt{n}} + \frac{12\sqrt{\zeta}}{n} \log(M\sqrt{n}/a) \right) \\
&\leq \frac{12\sqrt{\zeta}}{n} + \frac{12\sqrt{\zeta} \log(nM/(3\sqrt{\zeta}))}{n} \\
&\leq \frac{24\sqrt{\zeta}\{\log(nM/(3\sqrt{\zeta})) \vee 1\}}{n}.
\end{aligned}$$

Using Lemma A.1 with $r = 2M^2$, we have with probability greater than $1 - \epsilon$,

$$\begin{aligned}
\mathbb{E}[\mathcal{L}(f(x), y)] &\leq \frac{1}{n} \sum_{i=1}^{n} \mathcal{L}\left( f(\mathbf{x}_i), y_i \right) + \frac{144\sqrt{\zeta}\{\log(nM/(3\sqrt{\zeta})) \vee 1\}}{n} \\
&\quad + \sqrt{\frac{4M^2 \log(2/\epsilon)}{n}} + \frac{32M \log(2/\epsilon)}{3n} \\
&= \frac{1}{n} \sum_{i=1}^{n} \mathcal{L}\left( f(\mathbf{x}_i), y_i \right) + \mathcal{O} \left( \frac{\tilde{\alpha}^{\frac{3}{2}} \sqrt{\ln(\tilde{d}^2 G)}}{n} \right)
\end{aligned}$$

for any $f \in \mathcal{H}$, which completes our proof. $\square$

## E.5 Other Theoretical Results

Based on Assumption 1, which states that the distribution $\mathbb{P}$ satisfies the sub-Gaussian property, we can bound $\|X\|_2$ in Eq. (6) with high probability, leading to more applicable bounds in Corollary E.1.

**Corollary E.1** (Sub-Gaussian Generalization Bounds for KANs). *Under Assumption 1, 2 and 3, let fixed Lipschitz activations $\Sigma = \{\sigma_k : k \in [G]\}$ and weight tensor $\mathcal{W}_1^L = \{\mathbf{W}_1, \mathbf{W}_2, \ldots, \mathbf{W}_L\}$ be given. Then for $(\boldsymbol{x}, y), (\boldsymbol{x}_1, y_1), \ldots, (\boldsymbol{x}_n, y_n)$ drawn iid from sub-Gaussian distribution $\mathbb{P}$, we have with probability greater than $1 - \epsilon - \delta$,*

$$R(F_{\mathcal{W}_1^L}) - \hat{R}(F_{\mathcal{W}_1^L}) \leq \mathcal{O}\left(\frac{KR_{\mathcal{W}_1^L}\max_i |\rho(y_i)|}{n}\log(\tilde{d}^2 G)\sqrt{\tilde{d}\log\left(\frac{1}{\delta}\right)} + \sqrt{\frac{1/\epsilon}{n}}\right), \qquad (21)$$

*where $K$ was defined in Assumption 1.*

*Proof.* We conclude the result immediately by plugging Eq. (2) into Eq. (6). $\qquad\square$

We introduce Lemma E.1 that ensures the Lipschitz property of the composition of Lipschitz functions.

**Lemma E.1.** *If $\sigma : \mathbb{R} \to \mathbb{R}$ and $l : \mathbb{R} \to \mathbb{R}$ are both Lipschitz continuous functions with Lipschitz norms $\|\sigma\|_{Lip}$ and $\|\ell\|_{Lip}$, respectively, then the composition $l \circ \sigma(x)$ is Lipschitz continuous with Lipschitz norm $\|\ell \circ \sigma\|_{Lip} \leq \|\ell\|_{Lip}\|\sigma\|_{Lip}$.*

The proof of Lemma E.1 is provided in Appendix C.3. Lemma E.1 demonstrates that composing with a univariate function having a small Lipschitz norm reduces the Lipschitz norm of the activations family, thereby reducing the complexity of the functions represented by LipKANs.

**Corollary 5.1** (Generalization Bounds for LipKANs) Under Assumption 1, 2, and 3, let fixed Lipschitz activations $\Sigma = \{\sigma_k : k \in [G]\}$, weight tensor $\mathcal{W}_1^L = \{\mathbf{W}_1, \mathbf{W}_2, \ldots, \mathbf{W}_L\}$, and Lipschitz function $l : \mathbb{R} \to \mathbb{R}$ with $\|l\|_{\text{Lip}} \leq 1$. Then for $(\boldsymbol{x}, y), (\boldsymbol{x}_1, y_1), \ldots, (\boldsymbol{x}_n, y_n)$ drawn i.i.d. from sub-Gaussian distribution $\mathbb{P}$, we have with probability greater than $1 - \epsilon - \delta$,

$$R(F_{\mathcal{W}_1^L, l}) - \hat{R}(F_{\mathcal{W}_1^L, l}) \leq \mathcal{O}\left(\frac{KR_{\mathcal{W}_1^L, l}\max_i |\rho(y_i)|}{n}\log(\tilde{d}^2 G)\sqrt{\tilde{d}\log\left(\frac{1}{\delta}\right)} + \sqrt{\frac{1/\epsilon}{n}}\right),$$

where the right-hand side of the inequality is also bounded by the upper bound in Eq. (21).

*Proof.* By Lemma E.1,

$$R_{\mathcal{W}_1^L, l} = \left(\|\ell \circ \Sigma\|_{\text{Lip}}^L \prod_{i=1}^L \|\mathbf{W}_i\|_\sigma\right)\left(\sum_{i=1}^L \|\mathbf{W}_i^{\text{T}}\|_{2,2,1}^{2/3}\right)^{\frac{3}{2}}$$

$$\leq \left(\|\Sigma\|_{\text{Lip}}^L \prod_{i=1}^L \|\mathbf{W}_i\|_\sigma\right)\left(\sum_{i=1}^L \|\mathbf{W}_i^{\text{T}}\|_{2,2,1}^{2/3}\right)^{\frac{3}{2}},$$

where the right hand side of the inequality represents the complexity of KANs, thereby proving the conclusion. $\qquad\square$

# F    Limitations and Future Work

Our analysis currently uses the sub-Gaussian assumption to control input tail behavior and derive generalization bounds via standard concentration tools. However, the framework is not inherently restricted to sub-Gaussian inputs. In principle, it can extend to broader distribution classes—such as sub-exponential or heavy-tailed distributions—by leveraging tools from empirical process theory, including Bernstein-type inequalities, truncation methods, or Orlicz norm bounds. We view this as a natural direction for future theoretical extensions. For visualization of complexity metrics and empirical study of covering bounds, we leave these directions for future work, viewing them as promising avenues to further connect our theoretical analysis with practical model behavior.

