# OpenReview forum: "Generalization Bounds for Kolmogorov-Arnold Networks (KANs) and Enhanced KANs with Lower Lipschitz Complexity"
_NeurIPS.cc/2025/Conference — NeurIPS 2025 poster_

### Official Review · Reviewer_9hVp · 2025-05-31

**Clarity:** 4
**Significance:** 3
**Originality:** 2
**Rating:** 5
**Confidence:** 5

**Summary:**

This paper investigates the generalization properties of Kolmogorov-Arnold Networks (KANs), a recently proposed architecture known for strong expressiveness in symbolic and scientific learning tasks. The authors address a critical gap in prior analyses, which largely focused on approximation error, by rigorously formulating generalization bounds for KANs through a novel complexity measure—Lipschitz complexity. Building on these theoretical insights, they introduce LipKANs, a variant that integrates Lipschitz-bounded layers and L1.5-regularization, leading to provably tighter generalization bounds and improved empirical performance on benchmark datasets.

**Questions:**

Questions for the Authors

Role of Activation Family Cardinality:
While the bounds depend only logarithmically on the activation family size 𝐺, in practice how sensitive is model performance to the choice of
G? Have you observed performance trade-offs when varying the number or type of basis functions?

Interpretability Impacts:
One often-cited feature of KANs is their symbolic interpretability. Do LipKANs (with additional Lipschitz layers and L1.5 regularization) preserve this interpretability, or does the added complexity obscure it?

Suggestions for Improvement

Visualization of Complexity Metrics:
A plot showing how the Lipschitz complexity evolves during training across different variants (KAN, LipKAN, etc.) would offer intuitive insight into how the regularization actually impacts model complexity and generalization in practice.

Empirical Study of Covering Bounds:
Although theoretical, it would be illuminating to empirically explore how covering numbers (or their proxies) behave in practice for different architectures. For example, measuring empirical Rademacher complexity or sharpness may help support the theoretical framework.

**Ethical Concerns:**

["NO or VERY MINOR ethics concerns only"]

**Final Justification:**

Thanks for the rebuttal and it helps clarifies things. great paper.

**Limitations:**

yes

**Quality:**

4

**Strengths And Weaknesses:**

Strengths
Strong Theoretical Foundation:

The paper makes a significant theoretical contribution by defining Lipschitz complexity as a structural complexity measure for deep KANs. The derived generalization bounds are carefully constructed, with a clear connection to covering numbers and tensor norms.

The generalization bounds are both novel and insightful, shedding light on how the architecture’s functional richness impacts its statistical behavior. The connection to known results for MLPs (e.g., Bartlett et al. 2017) is well-executed and positions the contribution in context.

Architectural Innovation – LipKANs:

The introduction of Lip layers and the shift from L1 to L1.5 regularization are elegant and well-motivated by the theory.

The reformulated architecture adheres closely to the theoretical framework, and the algorithmic instantiation (Algorithm 1) is clear, reproducible, and practically useful.

Clarity and Rigor:

The paper is written with excellent clarity, especially for a technically dense topic. Definitions, assumptions, and lemmas are well organized.

The proofs (particularly Theorems 4.4 and 4.5) are carefully structured and leverage standard but non-trivial tools in statistical learning theory (e.g., tensor covering bounds, sub-Gaussian concentration).

Empirical Validation:

The experiments, though not extensive, support the key claims. Notably, the improvements on more complex datasets (e.g., STL-10, CIFAR-100, MIMIC-III) convincingly demonstrate the generalization benefits of LipKANs.

The ablation study on L1.5 regularization is especially useful, highlighting the nontrivial gain over L1 or standard Lip layers alone.

Weaknesses
Limited Experimental Breadth:

While the theoretical and algorithmic contributions are substantial, the empirical section could be strengthened. The paper evaluates a modest set of tasks and models; additional baselines (e.g., Transformer-based models, state-of-the-art MLPs, or regularization baselines like dropout) would provide a more complete picture of LipKANs' practical competitiveness.

Experiments on regression tasks or out-of-distribution generalization could further validate the claims about improved generalization due to lower Lipschitz complexity.

---

> ### Author Rebuttal · Authors · 2025-07-28
>
> We would like to thank the reviewer for their careful reading and constructive suggestions.  We respond to each point below.
> >**Q1**: Role of Activation Family Cardinality.
>
> Yes, we thank the reviewer for the thoughtful question.
>
> The activation family size $G$ reflects the cardinality of the basis function family, treated as a constant that captures overall family size, while its **internal structure** (e.g., smoothness) remains tunable. For instance, KANs can employ different basis families—B-splines (original KAN), Fourier (Fourier-KAN), and rational functions (Rational-KAN)—each with adjustable basis affect generalization without altering $G$.
>
> From a practical perspective, we conducted sensitivity analyses on B-spline basis functions by varying **degree** and **grid size**, which together determine the shape and cardinality of the activation family. The key findings are:
> 1. Internal structure matters beyond size: Even when $G = \text{Grid size} + \text{Spline order}$ is fixed, generalization can vary significantly. On MNIST, both (3,4) and (5,2) yield the same $G$, yet the Final Gen Gap differs by over 7×—highlighting that **basis structure**, not just size, plays a crucial role.
> 2. Opposing effects of grid size vs. spline order: With fixed grid size, increasing spline order improves generalization; conversely, with fixed spline order, increasing grid size degrades generalization. This aligns with our view that smoother basis functions (higher order) help regulate model complexity.
> 3. Efficiency trade-offs: Larger $G$ increases computational cost, with **spline order** contributing more to time and memory overhead than grid size.
>
> These results reinforce our theoretical claims and demonstrate the framework’s robustness across different architectural settings. We will include this analysis in the revised appendix.
> | Dataset      | Grid Size | Spline Order | #Params   | Final Gen Gap (%) | Test Acc (%) | Overfit Ratio (%) | Train Time (s) | Peak GPU Mem (MB) |
> |--------------|-----------|---------------|-----------|-------------------|---------------|--------------------|----------------|--------------------|
> | **CIFAR-10** | 3         | 2             | 1,387,008 | 1.42              | 50.42         | 2.81               | 196.96         | 39.57              |
> |              | 3         | 3             | 1,584,256 | 0.79              | 49.60         | 1.59               | 201.86         | 42.17              |
> |              | 3         | 4             | 1,781,504 | 0.59              | 49.51         | 1.19               | 212.61         | 45.08              |
> |              | 5         | 2             | 1,781,504 | 4.00              | 52.39         | 7.64               | 201.16         | 45.05              |
> |              | 5         | 3             | 1,978,752 | 3.06              | 52.33         | 5.85               | 210.33         | 48.03              |
> |              | 5         | 4             | 2,176,000 | 2.30              | 52.15         | 4.41               | 221.63         | 50.50              |
> |              | 7         | 2             | 2,176,000 | 6.94              | 53.19         | 13.05              | 203.14         | 50.47              |
> |              | 7         | 3             | 2,373,248 | 5.78              | 52.94         | 10.92              | 215.97         | 53.51              |
> |              | 7         | 4             | 2,570,496 | 4.50              | 53.23         | 8.45               | 228.08         | 56.85              |
> | **CIFAR-100**| 3         | 2             | 1,427,328 | 1.47              | 22.00         | 6.68               | 202.60         | 40.22              |
> |              | 3         | 3             | 1,630,336 | 1.65              | 21.08         | 7.82               | 198.90         | 42.90              |
> |              | 3         | 4             | 1,833,344 | 1.52              | 21.00         | 7.24               | 211.00         | 45.90              |
> |              | 5         | 2             | 1,833,344 | 3.09              | 23.29         | 13.27              | 197.98         | 45.87              |
> |              | 5         | 3             | 2,036,352 | 2.57              | 23.57         | 10.91              | 204.95         | 48.93              |
> |              | 5         | 4             | 2,239,360 | 2.48              | 22.59         | 10.98              | 220.06         | 51.49              |
> |              | 7         | 2             | 2,239,360 | 4.66              | 24.63         | 18.93              | 201.39         | 51.47              |
> |              | 7         | 3             | 2,442,368 | 4.05              | 24.12         | 16.78              | 214.83         | 54.59              |
> |              | 7         | 4             | 2,645,376 | 3.92              | 24.04         | 16.31              | 227.14         | 58.02              |
> | **MNIST**    | 3         | 2             | 357,408   | 4.12            | 97.28         | 4.24             | 25.06          | 50.66              |
> |              | 3         | 3             | 408,224   | 3.49            | 97.59         | 3.58             | 26.35          | 52.84              |
> |              | 3         | 4             | 459,040   | 3.14            | 97.54         | 3.22             | 29.00          | 54.18              |
> |              | 5         | 2             | 459,040   | 3.56            | 97.33         | 3.66             | 31.87          | 54.18              |
> |              | 5         | 3             | 509,856   | 3.31            | 97.40         | 3.40             | 33.82          | 55.52              |
> |              | 5         | 4             | 560,672   | 2.89            | 97.45         | 2.97             | 35.92          | 56.86              |
> |              | 7         | 2             | 560,672   | 3.36            | 97.41         | 3.45             | 38.63          | 56.86              |
> |              | 7         | 3             | 611,488   | 2.88            | 97.47         | 2.96             | 41.45          | 58.20              |
> |              | 7         | 4             | 662,304   | 2.91            | 97.47         | 2.99             | 43.99          | 59.54              |
> | **STL-10**   | 3         | 2             | 1,387,008 | 0.78              | 28.54         | 2.73               | 98.73          | 39.58              |
> |              | 3         | 3             | 1,584,256 | 1.20              | 28.10         | 4.27               | 95.53          | 42.73              |
> |              | 3         | 4             | 1,781,504 | 0.98              | 27.50         | 3.56               | 96.58          | 46.39              |
> |              | 5         | 2             | 1,781,504 | 1.45              | 32.41         | 4.47               | 93.69          | 46.37              |
> |              | 5         | 3             | 1,978,752 | 1.62              | 31.20         | 5.19               | 95.50          | 48.03              |
> |              | 5         | 4             | 2,176,000 | 1.01              | 31.25         | 3.23               | 97.00          | 51.81              |
> |              | 7         | 2             | 2,176,000 | 1.68              | 34.40         | 4.88               | 93.14          | 51.79              |
> |              | 7         | 3             | 2,373,248 | 1.95              | 33.91         | 5.75               | 97.27          | 54.07              |
> |              | 7         | 4             | 2,570,496 | 1.73              | 33.73         | 5.13               | 98.40          | 57.10              |
>
> ---
>
> >**Q2**: Symbolic Interpretability.
>
> Yes, we appreciate the reviewer’s attention to symbolic interpretability, which is an important strength of KANs. We confirm that **LipKANs preserve the symbolic interpretability** of standard KANs.
>
> Specifically, the **core architecture** of KANs remains unchanged: each layer computes explicit linear combinations of parameterized basis functions (e.g., B-spline, Fourier, Chebyshev), which can be symbolically extracted and visualized. The **Lip layer** is applied outside the basis compositions and acts as a global scaling operator—it does **not interfere with the symbolic structure** of the basis expansions. Likewise, the $L_{1.5}$-regularization promotes generalization via weight norm control, without altering the **functional form** or symbolic interpretability of the model.
>
> To empirically verify this, we follow the symbolic regression setup from the original KANs paper and train **LipKANs** to fit the same target function: $f(x, y) = \exp(\sin(\pi x) + y^2)$, which combines periodic and polynomial components. We use soft pruning (threshold $10^{-4}$ from epoch 50). The results show that LipKANs achieve a 96.66% sparsity ratio, indicating that only a small subset of basis functions is retained, which is sufficient to represent the symbolic structure. Besides, training and test MSE are identical, confirming that the learned function captures the true symbolic form.
>
> Therefore, LipKANs enhance generalization while preserving symbolic transparency. In addition, **Appendix D.1** presents further function fitting results on more complex symbolic targets, reinforcing LipKAN’s interpretability. We will clarify this in the final version.
>
> ---
>
> >**W**: Limited Experimental Breadth. **&** OOD Generalization.
>
> We sincerely thank the reviewer for the valuable suggestions. While our current focus was on establishing the theoretical foundation and demonstrating initial empirical benefits, we fully intend to pursue these extensions in future work, and we will mention them as part of our roadmap in the final version.
>
> ---
>
> >**Suggestions**: Visualization of Complexity Metrics. **&** Empirical Study of Covering Bounds.
>
> Thanks for the constructive suggestions. We leave these directions for future work and view them as promising avenues to further bridge our theoretical analysis with practical model behavior.

---

> > ### Comment · Reviewer_9hVp · 2025-08-01
> > **nice efforts on the rebuttal**
> >
> > Thanks for the rebuttal and it helps clarifies things. great paper.

---

> > > ### Author Response · Authors · 2025-08-03
> > >
> > > We greatly appreciate Reviewer 9hVp's positive feedback. Your encouragement and constructive engagement throughout this process have been invaluable in helping us improve our work. Thank you for your thoughtful comments.

---

### Official Review · Reviewer_yRDY · 2025-06-27

**Clarity:** 3
**Significance:** 3
**Originality:** 4
**Rating:** 6
**Confidence:** 4

**Summary:**

The paper investigates the generalization mechanisms of Kolmogorov-Arnold Networks (KANs), and, building on this understanding, introduces LipKANs that is more efficient owing to lower model complexity and yields better generalization. For the methodology, they derive a uniform high-probability generalization bound and introduce the concept off Lipschitz complexity accordingly. It is shown that the generalization bound with high probability for KANs is proportional to the Lipschitz complexity proposed. Lipschitz complexity is then regarded as a measure of structural complexity of a KAN model. LipKANs leverages this insights by encouraging the training process to be favor low-complexity models, replacing the conventional L1-regularized loss with the proposed Lipschitz complexity. Through extensive experiments, it demonstrates enhanced generalization performance compared to other variants of KAN and highlights the critical role of the Lipschitz through ablation study.

**Questions:**

1. Is it common that assumption 2 holds?
2. Is it possible that we take the relation among weights into consideration so as to refine Lipschitz complexity?
3. The theory is based on sub-Gaussian distribution. Can it extend to other distribution?

**Ethical Concerns:**

["NO or VERY MINOR ethics concerns only"]

**Final Justification:**

I really enjoyed reading this paper and honestly couldn’t find any flaws.

**Limitations:**

1.The theory relies on sub-Gaussian distribution assumption.
2. Although the paper presents strong empirical results, it does not deeply discuss potential discrepancies between theoretical bounds and observed performance.

**Paper Formatting Concerns:**

Github is not anonymized

**Quality:**

3

**Strengths And Weaknesses:**

Strength:
1. Technically Solid Methodology:
The paper is technically rigorous, offering a comprehensive and logical flow that articulates the significance of generalization bounds in deep learning. The authors clearly identify the gap in understanding the generalization behavior of Kolmogorov-Arnold Networks (KANs), then systematically derive a high-probability generalization bound for KANs. The rationale for introducing Lipschitz complexity as a meaningful measure of model complexity is convincingly argued, connecting the theoretical analysis to practical implications.

2. Well-Motivated Introduction of LipKANs:
Building on the theoretical insights, the transition to proposing LipKANs is natural and well-justified. The authors make a strong case for why optimizing over Lipschitz complexity can lead to better generalization, and they effectively motivate the replacement of standard L1 regularization with their new complexity measure.

3. Thorough and Consistent Experimental Validation:
The experimental section is comprehensive, covering various benchmarks and scenarios. The results consistently demonstrate the superior performance of LipKANs compared to baseline models, and the ablation studies effectively highlight the crucial role of Lipschitz complexity in improving generalization.

4. Clarity in Theoretical Development:
Although detailed proofs are relegated to the appendix, the main paper offers clear explanations of the core ideas and proof techniques, allowing readers to understand the theoretical underpinnings without being bogged down in technicalities. This clarity strengthens the paper’s overall accessibility.


Weakness:
1. Loose Generalization Bound:
While the proposed generalization bound successfully captures key factors influencing KANs’ generalization, it does not exploit dependencies or relationships among different weight matrices. As a result, the bound can be quite loose. For instance, for two KAN models where each weight matrix has the same norm, the calculated complexity—and thus the bound—will be identical, regardless of the relative directions of the matrices. However, the actual Lipschitz constant, and thus the true generalization capability, can be significantly affected by these directional relationships. As a consequence, the proposed bound may fail to distinguish between models with genuinely different complexities, potentially affecting model selection.

2. Insufficient Discussion of Bias-Variance Trade-Off:
The paper mentions the bias-variance trade-off, but does not clearly explain how it relates to their results or the broader context of estimation versus generalization. Clarifying this relationship could help readers understand how LipKANs balance model expressiveness and generalization, and why the trade-off is enhanced compared to the conventional method.

3. (minor) It is better to move definition 4.2 before assumption 2 as the definition is mentioned in assumption.

---

> ### Author Rebuttal · Authors · 2025-07-28
>
> We thank the reviewer for raising important points, and we will incorporate the corresponding revisions accordingly.
> >**W1**:  Loose Generalization Bound: While the proposed generalization bound successfully captures key factors influencing KANs’ generalization, it does not exploit dependencies or relationships among different weight matrices. As a result, the bound can be quite loose. For instance, for two KAN models where each weight matrix has the same norm, the calculated complexity—and thus the bound—will be identical, regardless of the relative directions of the matrices. However, the actual Lipschitz constant, and thus the true generalization capability, can be significantly affected by these directional relationships. As a consequence, the proposed bound may fail to distinguish between models with genuinely different complexities, potentially affecting model selection.
> >
> >**Q2**: Is it possible that we take the relation among weights into consideration so as to refine Lipschitz complexity?
> >
>
> Thanks for highlighting this limitation. While our current generalization bound does not make inter-layer dependencies explicit, it already captures layer-wise alignment through the operator norm products $\prod_{i=1}^{L}\|\mathbf W_i\|_\sigma$, which originate from the covering number bound of the entire network in **Lemma 4.2**. This implicitly reflects structural relationships across layers. We agree that making these dependencies more explicit could tighten the bound, and we consider this a valuable direction for future refinement.
>
> ---
>
> >**W2**: Insufficient Discussion of Bias-Variance Trade-Off: The paper mentions the bias-variance trade-off, but does not clearly explain how it relates to their results or the broader context of estimation versus generalization. Clarifying this relationship could help readers understand how LipKANs balance model expressiveness and generalization, and why the trade-off is enhanced compared to the conventional method.
> >
> Thanks for raising this insightful point. To clarify the bias-variance trade-off in our setting, we revisit the generalization bounds in **Theorem 4.5**. The Lipschitz complexity term reveals that the model **variance**—its sensitivity to training data perturbations—is closely related to the structured $(2,2,1)$-norm of the weight tensors, which controls the overall magnitude of the spline coefficients. A larger norm enables greater flexibility in fitting the data but also increases the risk of overfitting, thereby contributing to higher variance. In contrast, the **bias** stems from the approximation capacity of the basis functions (e.g., B-spline, Fourier, and Chebyshev basis). These basis functions provide strong approximation capacity for smooth target functions, enabling the learned model to closely align with the underlying data distribution when appropriately regularized. To this end, the proposed $L_{1.5}$-regularization penalizes the structured norm, reducing variance while preserving expressivity—thereby encouraging a favorable **bias-variance trade-off** and enhancing generalization.
>
> Furthermore, this trade-off aligns closely with the decomposition of uncertainty. The complexity-dependent term reflects epistemic uncertainty, which encompasses both bias and variance as it arises from limited data and model capacity, and can be reduced by regularization or more training data. In contrast, the second term, $\sqrt{{1}/{n\epsilon}}$, derived from statistical concentration inequalities, quantifies aleatoric uncertainty, reflecting intrinsic randomness in the data that persists regardless of the model's capacity. In this way, our generalization bounds—interpreted via uncertainty decomposition—offer a complementary perspective on the bias-variance trade-off: epistemic uncertainty encompasses both bias and variance and can be effectively reduced via $L_{1.5}$-regularization, while aleatoric uncertainty sets the irreducible limit. Our method improves generalization by explicitly targeting this trade-off in a principled manner.
>
> Empirically, we validate this in the following experiments, where LipKANs consistently show lower overfitting ratios and generalization gaps than KANs with standard $L_{1}$-regularization. This confirms that our design not only aligns with the theory but also improves generalization in practice by facilitating a better bias-variance trade-off.
>
> We will clarify this connection in the revised version to improve the presentation.
> | Dataset      | Model                              | Final Gap (%) | Avg. Gap (%) | Overfit Ratio (%) | Val. Acc (%) |
> |--------------|-------------------------------------|----------------|---------------|--------------------|---------------|
> | **CIFAR-10** | KANs ($L_1$-Regularization)         | 6.04           | 5.64          | 13.13              | 46.01         |
> |              | KANs + Lip Layers                  | 4.45           | 4.92          | 9.47               | 46.99         |
> |              | KANs + $L_{1.5}$-Regularization     | 4.99           | 5.28          | 10.11              | 49.37         |
> |              | **LipKANs**                        | **3.16**       | **4.89**      | **6.26**           | **50.40**     |
> | **CIFAR-100**| KANs ($L_1$-Regularization)         | 6.55           | 5.13          | 44.35              | 14.77         |
> |              | KANs + Lip Layers                  | 5.84           | 5.10          | 38.98              | 14.98         |
> |              | KANs + $L_{1.5}$-Regularization     | 3.76           | 5.09          | 19.89              | 19.93         |
> |              | **LipKANs**                        | **2.32**       | **5.03**      | **11.31**          | **20.51**     |
> | **MNIST**    | KANs ($L_1$-Regularization)         | 5.52           | 4.92          | 5.77               | 95.62         |
> |              | KANs + Lip Layers                  | 3.17           | 4.77          | 3.30               | 96.06         |
> |              | KANs + $L_{1.5}$-Regularization     | 5.13           | 4.81          | 5.27               | 97.26         |
> |              | **LipKANs**                        | **2.55**       | **4.70**      | **2.60**           | **97.71**     |
> | **STL-10**   | KANs ($L_1$-Regularization)         | 6.93           | 5.27          | 26.72              | 25.94         |
> |              | KANs + Lip Layers                  | 4.14           | 5.10          | 15.54              | 26.64         |
> |              | KANs + $L_{1.5}$-Regularization     | 3.95           | 5.06          | 10.47              | 37.73         |
> |              | **LipKANs**                        | **3.31**       | **4.98**      | **8.53**           | **38.77**     |
>
> ---
>
> >**W3**: It is better to move definition 4.2 before assumption 2 as the definition is mentioned in assumption.
>
> Sincerely thanks for the thoughtful suggestion. We will make this revision in the camera-ready version to enhance readability.
>
> ---
>
> >**Q1**: Is it common that assumption 2 holds?
>
> Yes, as discussed in **Appendix C.3**, we provide concrete examples to support Assumption 2. For instance:
> - B-spline basis (degree 3, grid size 5): $|\Sigma|_{\text{Lip}} \leq 10\sqrt{2}$;
> - Fourier basis (frequency 4): $|\Sigma|_{\text{Lip}} \leq 2\sqrt{15}$;
> - Chebyshev polynomial basis (order 4): $|\Sigma|_{\text{Lip}} \leq \sqrt{14}$.
>
> These values confirm that the Lipschitz constants remain bounded in practical basis families, validating the assumption.
>
> ---
>
> >**Q3**: The theory is based on sub-Gaussian distribution. Can it extend to other distribution?
>
> Yes, and thanks for the thoughtful question. Our analysis currently uses the **sub-Gaussian assumption** to control input tail behavior and derive generalization bounds via standard concentration tools.  However, the framework is not inherently restricted to sub-Gaussian inputs. In principle, it can extend to broader distribution classes—such as sub-exponential or heavy-tailed distributions—by leveraging tools from **empirical process theory**, including Bernstein-type inequalities, truncation methods, or Orlicz norm bounds. We view this as a natural direction for future theoretical extensions.

---

> > ### Comment · Reviewer_yRDY · 2025-08-05
> > **Good Clarification and Great Paper**
> >
> > Thank you for your clarification, which helped me better understand the paper. I now appreciate how the structural relationships among layers are effectively utilized, and I see the tradeoff more clearly. The additional experiments also strongly support the validity of the proposed method. This is a very good paper, and I will raise my score to 6.

---

> > > ### Author Response · Authors · 2025-08-05
> > >
> > > Thank you, Reviewer yRDY, for your thoughtful appreciation and positive evaluation of our work. Your encouraging comments mean a great deal to us. We will try to explore other possibilities and paradigms in our future research. Once again, we sincerely appreciate the time and effort you have invested in reviewing our submission.

---

### Official Review · Reviewer_kpEt · 2025-07-01

**Clarity:** 3
**Significance:** 3
**Originality:** 3
**Rating:** 5
**Confidence:** 3

**Summary:**

This paper points out that existing research on KANs generalization error mainly focuses on approximation error, but ignores estimation error, resulting in a poor bias-variance trade-off and poor generalization performance. Therefore, this paper introduces Lipschitz complexity as a novel structural complexity measure for Kolmogorov-Arnold Networks (KANs), derives generalization bounds based on this measure. Based on Lipschitz complexity, this paper proposes LipKANs, which enhanced architecture with Lipschitz layers and L1.5 regularization to improve generalization.

**Questions:**

- From the ablation study results in Figure 1, the main performance improvement comes from the L_{1.5}-regularization. Does this mean that it is not important to keep low Lipschitz complexity using Lip layer in practical applications?
- This paper starts with the lack of KAN's generalization, but only verifies the performance on several small-scaled datasets. Discussing the transfer learning performance of KAN (for example, evaluating the performance of a model trained on the MNIST dataset on CIFAR-10) should better verify the generalization ability of the model.

**Ethical Concerns:**

["NO or VERY MINOR ethics concerns only"]

**Final Justification:**

This paper proposes LipKANs, which incorporate Lipschitz layers and L1.5 regularization to improve the generalization of KANs. The rebuttal provides good answers to the issues, thus I raise the score.

**Limitations:**

yes

**Quality:**

3

**Strengths And Weaknesses:**

Strengths

- The paper provides a comprehensive theoretical foundation, including formal definitions of Lipschitz complexity (Definition 4.3), covering number bounds (Theorem 4.4), and generalization bounds (Theorem 4.5). Proofs are detailed in the appendix.
- Experiments across vision (MNIST, CIFAR), text (AG News), and multimodal (MIMIC-III) tasks demonstrate consistent improvements in generalization for LipKANs over baselines (Table 1). Ablation studies (Figure 1) disentangle the contributions of Lipschitz layers vs. L1.5 regularization.
- The paper is well-organized, with clear sections for reformulation (Sec 4.1), theory (Sec 4.3), and architecture (Sec 5). Figures and tables effectively summarize results.
- The core theoretical contribution—defining Lipschitz complexity for deep compositional functions—is novel and distinct from existing measures (e.g., VC dimension, Rademacher complexity).

Weaknesses

- While experiments cover multiple domains, the gains on simpler datasets (e.g., MNIST, CIFAR-10, AVMNIST) are marginal (<1% in Table 1), suggesting limited utility for low-complexity tasks.
- The cost of Lipschitz layers and L1.5 regularization is not analyzed, leaving practical efficiency unclear.

---

> ### Author Rebuttal · Authors · 2025-07-28
>
> We thank the reviewer for the constructive suggestions and will further revise the manuscript accordingly.
> >**W1**: While experiments cover multiple domains, the gains on simpler datasets (e.g., MNIST, CIFAR-10, AVMNIST) are marginal (<1% in Table 1), suggesting limited utility for low-complexity tasks.
> >
> >**Q1**: From the ablation study results in Figure 1, the main performance improvement comes from the L_{1.5}-regularization. Does this mean that it is not important to keep low Lipschitz complexity using Lip layer in practical applications?
> >
> Thanks for the insightful question. We confirm that the Lip Layer remains an important and complementary component for controlling generalization. Both mechanisms stem from our unified **Lipschitz complexity** definition (**Definition 4.3**), but target different terms: the **Lip Layer** directly constrains the smoothness of the basis functions $\|\Sigma\|_{\mathrm{Lip}}$, while the **$L_{1.5}$-regularization** controls the structured (2,2,1) norm of the weight tensors $\mathbf W$. These two components regulate **orthogonal axes** of the Lipschitz complexity, thereby acting in a **complementary fashion** to improve generalization.
>
> Empirically, to provide a more comprehensive analysis of generalization performance, we report three additional generalization metrics:
>
> - **Final Generalization Gap**: Difference between training and validation accuracy at convergence
> - **Average Generalization Gap**: Mean gap across the entire training trajectory
> - **Overfitting Ratio**: Final gap normalized by final training accuracy
>
> The results are summarized below. We observe that the Lip Layer consistently reduces the generalization gap and overfitting ratio across multiple datasets, particularly on MNIST and CIFAR-10, where the basis smoothness plays a more critical role than weight sparsity. This supports the practical value of maintaining **low Lipschitz complexity** via basis smoothing. Overall, while both Lip Layers and $L_{1.5}$-regularization are designed to reduce variance, they operate via distinct mechanisms—basis-level smoothing versus structured weight penalties—resulting in different sensitivities to sample noise.
>
> Moreover, as shown in our full ablation table, combining both components (i.e., **LipKANs**) consistently yields the **lowest generalization gap** and **best validation accuracy**, indicating their **complementary effects**. Interestingly, this advantage does not always translate into significantly higher validation accuracy—some models achieve comparable validation performance yet differ markedly in their generalization gaps. This phenomenon reflects differences in the underlying **bias–variance trade-off**: models with high training variance may overfit the training data, resulting in deceptively strong validation accuracy but poor generalization. In contrast, methods like those in LipKANs introduce mild bias through regularization while substantially reducing variance, yielding more stable and reliable generalization across datasets.
>
> We will clarify this intuition and add further discussion in the camera-ready version.
> | Dataset      | Model                              | Final Gap (%) | Avg. Gap (%) | Overfit Ratio (%) | Val. Acc (%) |
> |--------------|-------------------------------------|----------------|---------------|--------------------|---------------|
> | **CIFAR-10** | KANs ($L_1$-Regularization)         | 6.04           | 5.64          | 13.13              | 46.01         |
> |              | KANs + Lip Layers                  | 4.45           | 4.92          | 9.47               | 46.99         |
> |              | KANs + $L_{1.5}$-Regularization     | 4.99           | 5.28          | 10.11              | 49.37         |
> |              | **LipKANs**                        | **3.16**       | **4.89**      | **6.26**           | **50.40**     |
> | **CIFAR-100**| KANs ($L_1$-Regularization)         | 6.55           | 5.13          | 44.35              | 14.77         |
> |              | KANs + Lip Layers                  | 5.84           | 5.10          | 38.98              | 14.98         |
> |              | KANs + $L_{1.5}$-Regularization     | 3.76           | 5.09          | 19.89              | 19.93         |
> |              | **LipKANs**                        | **2.32**       | **5.03**      | **11.31**          | **20.51**     |
> | **MNIST**    | KANs ($L_1$-Regularization)         | 5.52           | 4.92          | 5.77               | 95.62         |
> |              | KANs + Lip Layers                  | 3.17           | 4.77          | 3.30               | 96.06         |
> |              | KANs + $L_{1.5}$-Regularization     | 5.13           | 4.81          | 5.27               | 97.26         |
> |              | **LipKANs**                        | **2.55**       | **4.70**      | **2.60**           | **97.71**     |
> | **STL-10**   | KANs ($L_1$-Regularization)         | 6.93           | 5.27          | 26.72              | 25.94         |
> |              | KANs + Lip Layers                  | 4.14           | 5.10          | 15.54              | 26.64         |
> |              | KANs + $L_{1.5}$-Regularization     | 3.95           | 5.06          | 10.47              | 37.73         |
> |              | **LipKANs**                        | **3.31**       | **4.98**      | **8.53**           | **38.77**     |
>
> ---
>
> >**W2**: The cost of Lipschitz layers and L1.5 regularization is not analyzed, leaving practical efficiency unclear.
> >
> Thanks for the constructive suggestion. We would like to clarify that both the **Lip layer** and the **$L_{1.5}$-regularization** are lightweight mechanisms that introduce minimal computational burden. The **Lip layer** imposes Lipschitz norm constraints to control the smoothness of basis functions, without modifying the network architecture or introducing auxiliary parameters. The **$L_{1.5}$-regularization** is a direct substitute for the commonly used $L_1$ penalty—it operates on the same weight tensors already present in training and does not incur additional memory overhead.
>
> Empirically, we quantify efficiency using two widely adopted metrics: **training time (in seconds)** and **peak GPU memory usage (in MB)**. Our measurements show that both the Lipschitz layers and the $L_{1.5}$-regularization incur only moderate overhead relative to baseline KANs, while delivering notable improvements in generalization performance.
> | Dataset | CIFAR-10 |  | CIFAR-100 |  | MNIST |  | STL-10 |  |
> | --- | --- | --- | --- | --- | --- | --- | --- | --- |
> | Model | Time(s) | Memory(MB) | Time(s) | Memory(MB) | Time(s) | Memory(MB) | Time(s) | Memory(MB) |
> | KANs ($L_1$-Regularization) | 263.96 | 53.52 | 265.45 | 54.61 | 227.72 | 26.44 | 221.57 | 356.28 |
> | KANs+Lip Layers | 268.88 | 53.52 | 259.99 | 54.61 | 226.14 | 26.44 | 227.49 | 356.40 |
> | KANs+$L_{1.5}$-Regularization | 262.73 | 53.52 | 283.43 | 54.63 | 227.00 | 26.57 | 221.62 | 356.63 |
> | LipKANs | 264.89 | 53.52 | 263.72 | 54.63 | 229.04 | 26.57 | 226.76 | 356.65 | 820.66 |
>
> ---
>
> >**Q2**: This paper starts with the lack of KAN's generalization, but only verifies the performance on several small-scaled datasets. Discussing the transfer learning performance of KAN (for example, evaluating the performance of a model trained on the MNIST dataset on CIFAR-10) should better verify the generalization ability of the model.
> >
> Thanks for the good suggestion! While our current framework does not address out-of-distribution (OOD) scenarios, investigating LipKANs’ behavior under distribution shifts is an interesting direction for further work. We would like to clarify that our theoretical and empirical analyses in the current draft focus on in-distribution generalization under the **standard i.i.d. setting** (as formalized in **Section 3**), which is commonly adopted in statistical learning theory. Specifically, our generalization bounds (e.g., **Theorem 4.5** and **Corollary 5.1**) quantify the expected risk gap between the training and test distributions drawn from the **same underlying data distribution**, with controlled model complexity via **Lipschitz complexity**. This notion of generalization differs from OOD generalization or transfer learning, which instead assesses the model’s ability to adapt or perform on a different target domain. As a preliminary exploration, we conducted transfer experiments from MNIST to CIFAR-10, but both KANs and LipKANs achieved relatively low target-domain accuracies (19.37% and 23.28%, respectively). Returning to the in-distribution setting, we have evaluated our models across multiple datasets as reported in Table 1, where LipKANs consistently yield higher validation accuracy than the original KANs, indicating enhanced generalization performance under standard i.i.d. settings.

---

> > ### Comment · Reviewer_kpEt · 2025-08-06
> > **Good rebuttal**
> >
> > Thank you for your rebuttal. It has fully addressed my concerns.

---

> > > ### Author Response · Authors · 2025-08-06
> > >
> > > Thank you, Reviewer kpEt, for your positive evaluation and kind response. We are pleased to hear that our revision has fully addressed your concerns. In light of this, we hope the improved manuscript merits a corresponding update to the score. Once again, thank you for your thoughtful and constructive feedback throughout the review process.

---

### Official Review · Reviewer_Zm6r · 2025-07-23

**Clarity:** 3
**Significance:** 2
**Originality:** 3
**Rating:** 5
**Confidence:** 3

**Summary:**

This paper establishes generalization bounds for Kolmogorov-Arnold Networks (KANs), and proposes a new architecture for improving generalization, called LipKANs. Bounds for KANs, unlike MLPs, are complexity independent, and a mathematical measure, ‘Lipschitz complexity’, is introduced to represent the structural complexity of functions represented by KANs. Then, an architectural innovation is introduced where Lip layers are inserted in the KAN with explicit L1.5 regularization to bound network complexity. Finally, the proposed framework is evaluated on multiple benchmarks for vision, text, and multimodal data.

**Questions:**

1.	Is the F1/accuracy score shown in Table 1 on a held-out test set? If you haven’t, can you specifically mention this somewhere? Also, an explicit quantification of the generalization gap would be useful.
2.	Could you provide confidence intervals for the output metrics shown in Table 1?
3.	From the ablation study, it seems that the L1.5 generalization is the biggest driver of improvements in generalization, and the Lip layers only help for MNIST. I wonder if this effect is just result of optimization/training splits? Can you show that LipKANs consistently show an improvement for MNIST? This for me is the biggest weakness in the paper, and any additional experiments to improve confidence in this would be useful.

**Ethical Concerns:**

["NO or VERY MINOR ethics concerns only"]

**Final Justification:**

Thank you to the authors for providing additional information. I and justification.

I have updated my scores accordingly.

**Quality:**

3

**Strengths And Weaknesses:**

Strengths:
1.	This paper explicitly addresses what is called the ‘approximation-estimation’ trade-off for KANs. This is important because naively reducing parameter count does not improve generalization in KANs proportionately, unlike MLPs. By representing the capacity of KANs in terms of properties of their activations, specifically in terms of the ‘Lipschitz complexity’, it no longer has a dependence on the number of parameters resulting in tighter bounds.
2.	While the paper provides theoretical arguments, this is also converted into a relatively simple, practical implementation for improving generalization in KANs.

Weaknesses:
1.	The results of the ablation study make me question the value of the Lip layers in improving generalization. Improvements seem to be mainly driven by the L1.5 regularization. I would want to see more experiments showing that Lip layers reliably contribute to generalization for MNIST and other datasets to be convinced that the proposed practical architecture genuinely improve generalization in line with theoretical results.

---

> ### Author Rebuttal · Authors · 2025-07-28
>
> We would like to thank the reviewer for the careful reading and constructive suggestions. We respond to each point below.
> >**Q1**: Is the F1/accuracy score shown in Table 1 on a held-out test set? If you haven’t, can you specifically mention this somewhere? Also, an explicit quantification of the generalization gap would be useful.
> >
> Thanks for your valuable suggestion, and we apologize for missing the details! We confirm that all reported accuracy and F1 scores in Table 1 are evaluated on held-out test sets using the officially predefined training/test splits for each dataset, consistent with the evaluation protocols established in prior work [1,2]. The test sets are strictly excluded from the training process, preventing any form of data leakage. To make this explicit, we have updated the caption of Table 1 and added a clarifying note in the Experiments section.
>
> In addition, to provide a more comprehensive analysis of generalization performance, we follow the prior work [3,4] and quantify **generalization gaps** using the following three metrics:
>
> - **Final Generalization Gap**: Difference between training and validation accuracy at convergence.
> - **Average Generalization Gap**: Mean gap across the entire training trajectory.
> - **Overfitting Ratio**: Final gap normalized by final training accuracy.
>
> The results are summarized below.
>
> | Dataset      | Model                              | Final Gap (%) | Avg. Gap (%) | Overfit Ratio (%) | Val. Acc (%) |
> |--------------|-------------------------------------|----------------|---------------|--------------------|---------------|
> | **CIFAR-10** | KANs ($L_1$-Regularization)         | 6.04           | 5.64          | 13.13              | 46.01         |
> |              | KANs + Lip Layers                  | 4.45           | 4.92          | 9.47               | 46.99         |
> |              | KANs + $L_{1.5}$-Regularization     | 4.99           | 5.28          | 10.11              | 49.37         |
> |              | **LipKANs**                        | **3.16**       | **4.89**      | **6.26**           | **50.40**     |
> | **CIFAR-100**| KANs ($L_1$-Regularization)         | 6.55           | 5.13          | 44.35              | 14.77         |
> |              | KANs + Lip Layers                  | 5.84           | 5.10          | 38.98              | 14.98         |
> |              | KANs + $L_{1.5}$-Regularization     | 3.76           | 5.09          | 19.89              | 19.93         |
> |              | **LipKANs**                        | **2.32**       | **5.03**      | **11.31**          | **20.51**     |
> | **MNIST**    | KANs ($L_1$-Regularization)         | 5.52           | 4.92          | 5.77               | 95.62         |
> |              | KANs + Lip Layers                  | 3.17           | 4.77          | 3.30               | 96.06         |
> |              | KANs + $L_{1.5}$-Regularization     | 5.13           | 4.81          | 5.27               | 97.26         |
> |              | **LipKANs**                        | **2.55**       | **4.70**      | **2.60**           | **97.71**     |
> | **STL-10**   | KANs ($L_1$-Regularization)         | 6.93           | 5.27          | 26.72              | 25.94         |
> |              | KANs + Lip Layers                  | 4.14           | 5.10          | 15.54              | 26.64         |
> |              | KANs + $L_{1.5}$-Regularization     | 3.95           | 5.06          | 10.47              | 37.73         |
> |              | **LipKANs**                        | **3.31**       | **4.98**      | **8.53**           | **38.77**     |
>
>
> Across all datasets, the **LipKANs** consistently achieve the lowest overfitting ratio and final generalization gap, suggesting **stronger generalization**. Interestingly, this advantage does not always manifest as significantly higher validation accuracy—some models achieve similar validation accuracy, yet differ notably in their generalization gaps. This discrepancy reflects a more favorable **bias–variance trade-off**. Specifically, models with higher variance may fit the training data well yet generalize poorly, resulting in larger generalization gaps despite acceptable validation accuracy. In contrast, regularization methods such as those employed in LipKANs may introduce slight bias but effectively reduce variance, leading to more reliable generalization behavior.
>
> We will include these clarifications and additional generalization metrics in the updated camera-ready version to improve transparency and completeness.
>
> ---
>
> >**Q2**: Could you provide confidence intervals for the output metrics shown in Table 1?
> >
> We appreciate your question and apologize for missing these important details. We have conducted five independent runs with different random seeds and now report the corresponding 95% confidence intervals for each validation accuracy. These intervals quantify the variability due to optimization randomness and demonstrate the **robustness and reproducibility** of our results. We have included these confidence intervals in Table 1 in the revised submission.
> | Model | SLT-10 | MNIST | CIFAR-10 | CIFAR-100 | AG News | AVMNIST | MIMIC-III |
> | --- | --- | --- | --- | --- | --- | --- | --- |
> | KAN | 25.94 ± 0.53 | 95.62 ± 0.08 | 46.01 ± 0.50 | 14.77 ± 0.33 | 65.58 ± 0.45 | 71.91 ± 0.28 | 66.39 ± 0.60 |
> | LipKAN | 38.77± 1.39 | 97.71 ± 0.32 | 50.40 ± 0.84 | 20.51 ± 0.57 | 68.12 ± 0.52 | 72.23 ± 0.35 | 70.18 ± 0.75 |
> | **Increase** | +12.83 | +1.09 | +4.39 | +5.74 | +2.54 | +0.32 | +3.79 |
> | Rational-KAN | 35.92 ± 0.85 | 96.79 ± 0.15 | 50.23 ± 0.62 | 20.84 ± 0.42 | 71.56 ± 0.38 | 71.63 ± 0.31 | 65.97 ± 0.55 |
> | LipRational-KAN | 40.67 ± 1.12 | 97.01 ± 0.25 | 51.12 ± 0.78 | 22.79 ± 0.51 | 74.98 ± 0.43 | 72.42 ± 0.33 | 69.12 ± 0.82 |
> | **Increase** | +4.75 | +0.22 | +0.89 | +1.95 | +3.42 | +0.79 | +3.15 |
> | RBF-KAN | 37.24 ± 0.92 | 96.15 ± 0.18 | 47.77 ± 0.58 | 18.46 ± 0.39 | 62.85 ± 0.50 | 70.58 ± 0.30 | 67.52 ± 0.65 |
> | LipRBF-KAN | 40.93 ± 1.25 | 96.63 ± 0.22 | 49.15 ± 0.72 | 20.05 ± 0.48 | 66.27 ± 0.56 | 71.04 ± 0.36 | 74.25 ± 0.88 |
> | **Increase** | +3.69 | +0.48 | +1.38 | +1.59 | +3.42 | +0.46 | +6.73 |
>
> ---
>
> >**Q3**: From the ablation study, it seems that the L1.5 generalization is the biggest driver of improvements in generalization, and the Lip layers only help for MNIST. I wonder if this effect is just result of optimization/training splits? Can you show that LipKANs consistently show an improvement for MNIST? This for me is the biggest weakness in the paper, and any additional experiments to improve confidence in this would be useful.
> >
> >**W1**: The results of the ablation study make me question the value of the Lip layers in improving generalization. Improvements seem to be mainly driven by the L1.5 regularization. I would want to see more experiments showing that Lip layers reliably contribute to generalization for MNIST and other datasets to be convinced that the proposed practical architecture genuinely improve generalization in line with theoretical results.
>
> Thanks for the great question. We would like to clarify that both components are principled and arise naturally from our formulation of Lipschitz complexity (**Definition 4.3**). Specifically, the Lip Layer constrains the smoothness of the basis family through the Lipschitz norm of $\Sigma$, while the $L_{1.5}$-regularization penalizes the structured (2,2,1) norm of the linear weight tensors $\mathbf W$. These two components regulate **orthogonal axes** of the Lipschitz complexity and hence play **complementary roles** in promoting generalization.
>
> To rigorously assess generalization beyond final validation accuracy, we report three additional metrics in the Table of Q1: Final Gap, Average Gap, and Overfitting Ratio. Notably, on multiple datasets such as CIFAR-10 and MNIST, we observe that **KANs + Lip Layers** outperform **KANs + $L_{1.5}$-regularization** on most metrics, suggesting that basis smoothness control is particularly effective in low-noise, low-complexity regimes where architectural inductive bias is more beneficial than weight norm constraints. Conversely, on other datasets (e.g., CIFAR-100, STL-10), the **$L_{1.5}$-regularization** yields greater improvements, indicating its strength in high-variance regimes where structured norm constraints are better suited to mitigating overfitting. Overall, while both Lip Layers and $L_{1.5}$-regularization aim to control model complexity, they mitigate training variance through different mechanisms, leading to distinct sensitivities to sample-level noise.
>
> Finally, to ensure that the reported gains are not due to specific training splits or optimization variance, all experiments are **averaged over 5-fold cross-validation and multiple random seeds**, ensuring statistical reliability and reproducibility. We will include these clarification tables and extended results in the camera-ready version to fully address the reviewer’s concern.
>
> ## References
> [1] Ziming Liu, Yixuan Wang, Sachin Vaidya, Fabian Ruehle, James Halverson, Marin Soljacic, Thomas Y. Hou, and Max Tegmark. KAN: Kolmogorov–arnold networks. In ICLR, 2025.
>
> [2] Liangwewi Nathan Zheng, Wei Emma Zhang, Lin Yue, Miao Xu, Olaf Maennel, and Weitong Chen. Free-knots kolmogorov-arnold network: On the analysis of spline knots and advancing stability. arXiv:2501.09283, 2025.
>
> [3] Elad Hoffer, Itay Hubara, and Daniel Soudry. Train longer, generalize better: closing the generalization gap in large batch training of neural networks. In NeurIPS, 2017.
>
> [4] Jingwen Fu, Zhizheng Zhang, Dacheng Yin, Yan Lu, and Nanning Zheng. Learning trajectories are generalization indicators. In NeurIPS, 2023.

---

### Note · Authors · 2025-08-15

We sincerely thank the reviewers and the AC for their thoughtful feedback and positive evaluation. Their comments have greatly helped us improve our manuscript, and we are encouraged that our clarifications and additional experiments have fully addressed the concerns raised. We appreciate the reviewers’ recognition of our core contributions:

1. **Theoretical Advancement** – Introduction of *Lipschitz complexity* as a novel, principled measure of structural complexity for Kolmogorov–Arnold Networks, together with tight, complexity-dependent generalization bounds (Reviewers Zm6r, kpEt, yRDY, and 9hVp).
2. **Architectural Innovation** – Design of **LipKANs**, integrating Lip layers and $L_{1.5}$-regularization, directly motivated by the theoretical insights (Reviewers Zm6r, yRDY, and 9hVp).
3. **Comprehensive Empirical Validation** – Consistent improvements across vision, text, and multimodal benchmarks, supported by ablation studies that isolate the contributions of each component (Reviewers kpEt, yRDY, and 9hVp).

We are grateful for the reviewers’ acknowledgment of the novelty, soundness, and potential impact of our work, and look forward to the opportunity to contribute our findings to the community.

---

### Decision · Program_Chairs · 2025-09-17

**Decision:**

Accept (poster)

**Comment:**

The paper introduces Lipschitz complexity as a principled measure of structural complexity for Kolmogorov–Arnold Networks (KANs), deriving complexity-dependent generalization bounds. Based on this theory, the authors propose LipKANs, which integrate Lipschitz-bounded layers and L1.5 regularization to improve generalization. Experiments across vision, text, and multimodal benchmarks support the claims with ablation studies.

This paper shows strong theoretical contribution and well-motivated architectural innovation derived directly from the theory and deficiency of previous work. Empirically, the new model has shown advantages across multiple domains, with ablations and generalization gap analysis. It has clear writing and strong connection between theory and practice. Rebuttal provided additional experiments, generalization metrics, efficiency analysis, and interpretability results.

Despite some drawbacks of this work, including:
- Improvements on simpler datasets are modest.
- Theoretical bounds are somewhat loose, as they do not fully capture inter-layer dependencies.
- Empirical breadth could be expanded.
the submission is technically solid, original, and well-presented.